# TAPVid-3D:
# A Benchmark for Tracking Any Point in 3D

**Skanda Koppula*[1,2], Ignacio Rocco*[1], Yi Yang[1], Joe Heyward[1],**
**João Carreira[1], Andrew Zisserman[1,3], Gabriel Brostow[2], Carl Doersch[1]**
[1]Google DeepMind    [2]University College London    [3]University of Oxford

## Abstract

We introduce a new benchmark, *TAPVid-3D*, for evaluating the task of long-range Tracking Any Point in 3D (TAP-3D). While point tracking in two dimensions (TAP-2D) has many benchmarks measuring performance on real-world videos, such as TAPVid-DAVIS, three-dimensional point tracking has none. To this end, leveraging existing footage, we build a new benchmark for 3D point tracking featuring 4,000+ real-world videos, composed of three different data sources spanning a variety of object types, motion patterns, and indoor and outdoor environments. To measure performance on the TAP-3D task, we formulate a collection of metrics that extend the Jaccard-based metric used in TAP-2D to handle the complexities of ambiguous depth scales across models, occlusions, and multi-track spatio-temporal smoothness. We manually verify a large sample of trajectories to ensure correct video annotations, and assess the current state of the TAP-3D task by constructing competitive baselines using existing tracking models. We anticipate this benchmark will serve as a guidepost to improve our ability to understand precise 3D motion and surface deformation from monocular video.

## 1   Introduction

For robots, humans, and other agents to effectively interact with the physical 3D world, it is necessary to understand a scene's structure and dynamics. This is a key ingredient to any general embodied intelligence: the ability to learn a world model to understand and predict the structure and motion of arbitrary scenes. It is attractive to leverage the vast amounts of monocular video data that is available cheaply, and use such signals to understand the geometry and 3D motion in real-world footage. But how well can current perception algorithms actually do this?

The field has seen many efforts to measure 3D and motion understanding from videos, each contributing a part of the overall goal. For example, monocular depth estimation is a widely recognized task [8, 36, 61]. However, success in depth estimation alone doesn't reveal whether the model understands how surfaces move from one frame to the next, and may not respect temporal continuity. For instance, a basketball spinning on its axis is often not visible in a sequence of depth maps. On the other end of the generality spectrum, 3D pose tracking, e.g. for rigid objects [15, 34] and people [13, 48], evaluates precise motion, but requires a known 3D model of each object, with articulations. Building parametric models and pose estimation methods for all animal classes is infeasible, much less for all the objects that a robot might encounter in, for example, an unseen, busy construction site.

An alternative and more general approach for 3D understanding observes that the world consists of particles, each of which individually follows a 3D trajectory through space. Measuring the motion of these points provides a way to measure 3D motion *without requiring any 3D shapes to be known*

---

*Equal contribution. Corresponding author: `skandak@google.com`

38th Conference on Neural Information Processing Systems (NeurIPS 2024) Track on Datasets and Benchmarks.

*a priori*. To this end, in TAPVid-3D, we focus on providing the community with a benchmark consisting of real world videos and three-dimensional point tracking annotations, spanning a wide variety of objects, scenes, and motion patterns.

Prior work on 2D understanding has shown that this kind of point-wise motion can be extremely valuable: both optical flow and the longer term two-dimensional Tracking-Any-Point (TAP) tasks have been applied to video editing [64], controllable video generation [57], robotic manipulation [53, 58], and more [9, 18]. TAP is an occlusion-aware, temporal extension of optical flow, which itself has a 3D extension called *scene flow* [37, 39, 40]. While useful, scene flow suffers from the same challenges as optical flow, namely, that it captures instantaneous motion and does not evaluate correct, occlusion-aware association of pixels over long sequences. A new three-dimensional TAP benchmark would provide a way to measure progress on many of these tasks: our TAPVid-3D benchmark targets evaluating general motion understanding, for models performing both dense and sparse particle tracking, in two and three dimensions.

Unfortunately, all the currently-used evaluations for TAP on real-world videos assess only 2D tracking ability (e.g. the TAP-Vid suite [10], BADJA [5], CroHD [51], and JHMDB [19]), and cannot evaluate the performance of 3D point tracking due to lack of access to the ground-truth metric position trajectories. While evaluations based on synthetic environments, like Kubric [14], RGB-Stacking [10], and Point Odyssey [68], could potentially provide 3D point tracking annotations, these introduce a significant domain gap with real-world scenes and are therefore less representative of model performance on real-world tasks.

Many applications stand to benefit from direct evaluation of three-dimensional point tracking capabilities and improvements to such models. For example, robotic manipulation tasks are likely to be easier with accurate 3D motion estimates, to understand the changing relative world position of the gripper, any objects, and the background. Video generation models would be more useful if creators were able to condition on exact motion tracks describing the 3D movement of both objects and the camera, as a director would do on a stage. Standard scene understanding tasks like velocity estimation, motion prediction, and object parts segmentation are simpler given the 3D motion tracks of individual points. Many visual odometry, mapping, and structure from motion pipelines rely on accurate 3D correspondences; with the ability to track 3D point correspondences from *any* pixel, such pipelines could be made more robust and accurate, even with many moving objects. Overall, the task of three dimensional point tracking provides a strict superset of information as compared to its 2D counterpart, is likely to be more useful in downstream applications, and provides a greater test of physical world motion understanding.

To this end, we introduce TAPVid-3D: a *real-world* benchmark for evaluating the Tracking Any Point in 3D (TAP-3D) task. The benchmark contributes: (1) a unification of three distinct real-world video data sources, with pipelines to generate, standardize, and validate consistent ground-truth $(x, y, z)$ 3D trajectories and occlusion information, (2) formalization of the TAP-3D task, with new metrics to measure accuracy of 3D track estimation, and (3) an assessment of the current state of TAP-3D, formed from the first real-world video evaluations of the nascent set of early 3D TAP models.

## 2   Related work

**Tracking Any Point.**   In recent years, long-range tracking of local image points has been formalized as the Tracking Any Point task (TAP) and evaluated using the, now standard, TAPVid benchmark [10], among others. Success on TAP consists of tracking the 2D trajectory of any given $(x, y)$ *query point*, defined at a particular frame $t$, throughout the rest of a given video. By definition, the query point is considered visible at the query frame, and associated to the material point of the scene which is observed at $(x, y, t)$. However, this material point may become occluded or go out of the image boundaries. To handle this, TAP models also must predict a binary visibility flag $v$ for each timestamp of the video. The tracked $(x, y)$ positions and visibility estimates are scored jointly using an all-encompassing average Jaccard metric. Most current TAP models [10–12, 17, 22] are limited to tracking in 2D pixel space. Recently, some works have started exploring the extension of the TAP problem to 3D (TAP-3D) [56, 60]. However, most TAP benchmarks containing real-world videos, such as TAPVid-DAVIS [10], Perception Test [42], CroHD [51] and BADJA [5], don't have 3D annotations, and therefore evaluations are still performed on the 2D tracking task. Concurrent to our work, Wang et al. [56] introduced both a synthetic (LSFOdyssey) and a real benchmark (LSFDriving)

for evaluating TAP-3D. However, their real benchmark is limited to the driving domain and only contains 180 test clips with 40 frames each. Our proposed TAPVid-3D benchmark is substantially larger and more diverse, containing 4000+ clips with durations between 25 and 300 frames.

**Scene flow estimation.** The scene flow estimation problem, introduced by Vedula et al. [54], seeks to obtain a dense, instantaneous, 3D motion field of a 3D scene, analogously to the way optical flow estimates a dense 2D motion field across consecutive frame pairs of a 2D video. The TAP-3D task is related to the scene flow problem in a similar way in which the TAP task is related to the optical flow problem. While scene flow seeks to obtain dense instantaneous motion estimation, TAP-3D seeks to obtain longer-range tracking, spanning tens or hundreds of frames. Furthermore, TAP-3D does not seek to produce dense fields of tracks, but is rather interested in tracking a sparse set of query points, which is more computationally tractable. From work in TAP-2D, we have observed that having motion representations with longer temporal context is useful for downstream tasks such as robotic manipulation, while having sparser spatial coverage is sufficient for many tracking applications.

**Pose estimation.** Closely related to point tracking is pose estimation and keypoint tracking. Many benchmarks have been proposed for 2D and 3D pose estimation [16, 27, 55, 66]. 3D pose estimation tasks and benchmarks largely focus on specific categories of moving objects, and for objects that are articulated: e.g. humans [20, 55], hands [52], animals [38, 65], and even jointed furniture [32]. For general motion and scene understanding, we aim to learn motion estimation across any object or scene pixel, expanding the generality of the task.

**Static scene reconstruction.** Static scene reconstruction, a fundamental problem in computer vision, has been advanced through techniques like Structure-from-Motion (SfM) and monocular depth estimation. Significant contributions include COLMAP [44] for state-of-the-art reconstructions and MVSNet [63], which enhances multi-view stereo depth estimation with deep learning. These studies collectively advance robust and precise static scene reconstruction. Evaluation of the local features and depth are crucial for static scene reconstruction methods. [45] provided a comparative evaluation of hand-crafted and learned features, while MegaDepth [28] improved single-view depth prediction using large scale multi-view Internet photo collections. Despite significant progress, static reconstruction struggles with dynamic environments, highlighting the need for dynamic scene methods.

**Dynamic scene reconstruction.** 3D reconstruction of dynamic scenes and objects is a widely studied problem in computer vision. Over the years, several methods have proposed solutions to this problem, starting with Non-rigid Structure-from-Motion methods (NRSfM) [2, 6]. While these methods have shown some success modelling simple motions like facial expressions and skeletal motions, they fail to generalize to arbitrary motions. More recently, deep-learning based methods have been used to perform 3D reconstruction of dynamic scenes. One line of works exploit Monocular Depth Estimation models [43] and performs test-time optimization on each given video to obtain smoother reconstructions, under the assumption that the frame rate of the camera is high with respect to the speed of the depicted object (quasi-static scene assumption) [26, 36, 67]. While these methods are more general than the classic NRSfM counterparts, they still require costly test-time optimization and fail to model rapid motions. Other lines of work attempt to fit a neural-scene representation to each video, such as neural-radiance fields [29, 30] or 3D Gaussian Splatting [62]. However, these methods require a costly per-video optimization, and typically make smoothness and local-rigidity assumptions about the motion of the points in the scene. We believe the development of TAP-3D methods should significantly help for the problem of dynamic scene reconstruction, as these models can run in a feed-forward manner, without requiring test-time optimization, and don't need any explicit motion prior assumptions as they can learn these from data.

Table 2 summarizes the focus areas of measurement for common scene understanding benchmarks. On the bottom row is the proposed TAP-3D task, and corresponding benchmark, TAPVid-3D, which brings to the table a more complete test of dynamic scene understanding in one simple evaluation.

## 3  TAPVid-3D

We build a real-world benchmark for evaluating Tracking Any Point in 3D (TAP-3D) models. To do this, we leveraged three publicly available datasets: (i) Aria Digital Twins [41], (ii) DriveTrack [3, 50]

| Benchmark Type | Long Term Continuity | 3D | Pixel Level Occlusion | Pixel Level Motion | Non-rigid Surfaces | No Prior 3D Model |
|---|---|---|---|---|---|---|
| Monodepth | ✗ | ✓ | ✗ | ✗ | ✓ | ✓ |
| 2D Point Tracking (TAP) | ✓ | ✗ | ✓ | ✓ | ✓ | ✓ |
| Scene flow | ✗ | ✓ | ✗ | ✓ | ✓ | ✓ |
| 3D Pose Tracking | ✓ | ✓ | ✗ | ✓ | ✗ | ✗ |
| 3D Object Box Tracking | ✓ | ✓ | ✗ | ✗ | ✗ | ✓ |
| **3D Point Tracking (TAP-3D)** | ✓ | ✓ | ✓ | ✓ | ✓ | ✓ |

Table 1: The proposed TAPVid-3D benchmark provides a unique set of characteristics, not covered in previous tasks or benchmarks. It extends the temporal continuity, occlusion modeling, and motion estimation capabilities of TAP benchmarks into 3D. We give examples of each type of benchmark in each row.

| Benchmark Type | Long Term Continuity | 3D | Pixel Level Occlusion | Pixel Level Motion | Non-rigid Surfaces | No Prior 3D Model |
|---|---|---|---|---|---|---|
| Monodepth [7, 28, 49] | ✗ | ✓ | ✗ | ✗ | ✓ | ✓ |
| 2D Point Tracking (TAP) [10] | ✓ | ✗ | ✓ | ✓ | ✓ | ✓ |
| Scene flow [24, 59] | ✗ | ✓ | ✗ | ✓ | ✓ | ✓ |
| 3D Pose Tracking [33, 55] | ✓ | ✓ | ✗ | ✓ | ✗ | ✗ |
| 3D Object Box Tracking [31, 50] | ✓ | ✓ | ✗ | ✗ | ✗ | ✓ |
| **3D Point Tracking (TAP-3D)** | ✓ | ✓ | ✓ | ✓ | ✓ | ✓ |

Table 2: The proposed TAPVid-3D benchmark provides a unique set of characteristics, not covered in previous tasks or benchmarks. It extends the temporal continuity, occlusion modeling, and motion estimation capabilities of TAP benchmarks into 3D. We give examples of each type of benchmark in each row.

and (iii) Panoptic Studio [21]. These data sources span different application domains, environments, and video characteristics, deriving ground truth tracking trajectories from different sensor types. For instance, Aria Digital Twins is a dataset of egocentric video, and is more close to bimanual robotic manipulation problems, where the camera is robot mounted and sees the actions from a first person view. DriveTrack features footage captured from a Waymo car navigating outdoor scenes, akin to tasks in robotic navigation and outdoor, rigid-body scene understanding tasks. Finally, Panoptic Studio captures third-person views of people performing diverse actions within an instrumented dome. It presents complex human movement which aligns more closely with NRSfM-adjacent tasks. We believe that, combined, these data sources present a diverse and comprehensive benchmark of TAP-3D capabilities for many potential downstream tasks. We describe our pipeline to extract ground truth metric 3D point trajectories from each source in the next sections, with samples in Figure 1.

Table 3 shows the summary statistics across the entire dataset, and for the dataset subdivisions corresponding to each constituent data source. As there are comparatively a large number of videos (for reference, the commonly used TAPVid-DAVIS in TAP-2D has 30 videos, whereas TAPVid-3D has two orders of magnitude more), we release two splits: a `minival` split, with 50 videos from each data source, and a `full_test` split, containing all videos in the benchmark, without the `minival` videos. The `minival` is intentionally lightweight, and likely most useful for online evaluation during training.

| Dataset split | #clips (minival) | #trajs per clip | #frames per clip | #videos | #scenes | resolution | fps |
|---|---|---|---|---|---|---|---|
| Aria Digital Twins | 1956 (50) | 1024 | 300 | 215 | 2 | $512 \times 512$ | 30 |
| DriveTrack | 2457 (50) | 256 | $25 - 300$ | 2457 | 252 | $1920 \times 1280$ | 10 |
| Panoptic Studio | 156 (50) | 50 | 150 | 156 | 1 | $640 \times 360$ | 30 |
| TAPVid-3D | 4569 (150) | $50 - 1024$ | $25 - 300$ | 2828 | 255 | Multiple | 10/30 |

Table 3: Overview statistics of the TAPVid-3D benchmark dataset, for all three constituent splits and in total. Clips in the benchmark are temporally sampled from their original video.

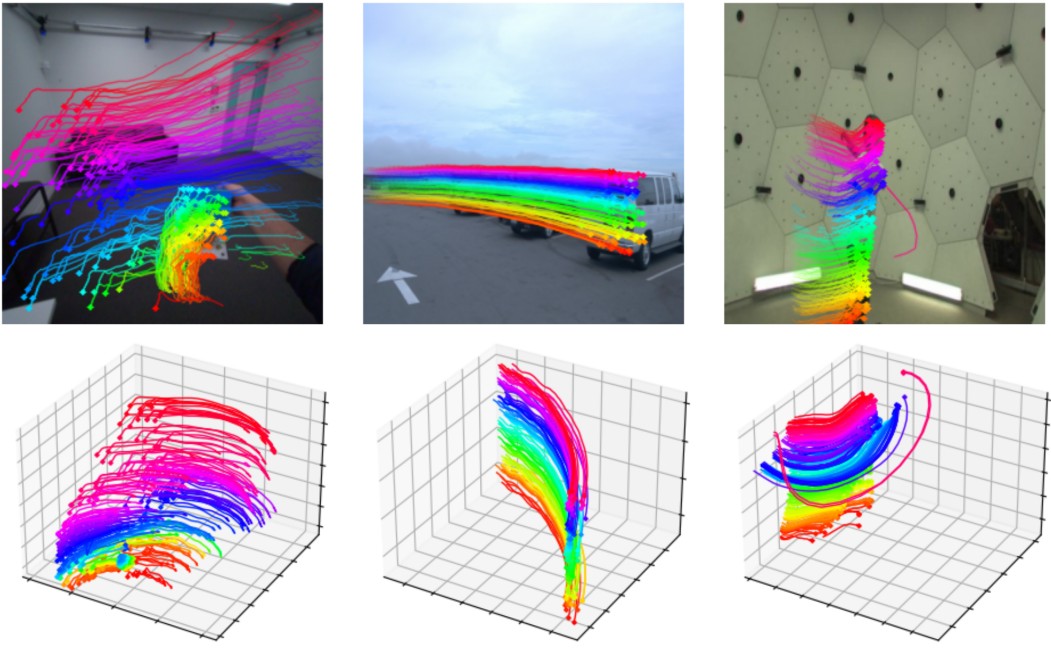

Figure 1: Random samples from TAPVid-3D: on the top row, we visualize the point trajectories projected into the 2D video frame; on the bottom row, we visualize the metric 3D point trajectories. From left to right, we show one example from each constituent data source: ADT, DriveTrack and Panoptic Studio.

## 3.1  Aria Digital Twins

This split employs real videos captured with the Aria glasses [1] inside different recording studios, which mimic household environments. Accurate digital replicas of these studios, created in 3D modeling software, are used to obtain pseudo-ground truth annotations of the original footage. This includes annotations such as segmentation masks, 3D object bounding boxes and depth maps. We leverage these annotations to extract 3D trajectories from given 3D query points. In particular, given a video $V = \{I_t\}_{t=1,\ldots,T}$ with $T$ frames and $W \times H$ spatial resolution, with corresponding object segmentation masks $\{S_t\} \subset \mathbb{Z}^{W \times H}$, and depth maps $\{D_t\} \subset \mathbb{R}^{W \times H}$, extrinsic world-to-camera poses* $\{(P_{cam}^w)_t\} \subset \mathbb{R}^{3 \times 4}$, camera intrinsics $K \in \mathbb{R}^{3 \times 3}$, and a query point $q = (x_q, y_q, t_q)$, we first extract the query point's 3D position $Q_{cam}^\dagger$ in the camera coordinate frame, with

$$(Q_{cam})_{t_q} = K^{-1}(x_q, y_q, 1)^T \cdot D_{t_q}(x_q, y_q). \tag{1}$$

Additionally, we obtain the object ID of the query point from the segmentation mask $q_{id} = S_{t_q}(x_q, y_q)$, and use it to retrieve the 3D object pose of the query object $(P_{obj}^w)_{t_q}$, which converts points from world coordinate frame to the object coordinate frame. This allows us to compute the query point position in the object's frame of reference as

$$Q_{obj} = (P_{obj}^w)_{t_q}(P_w^{cam})_{t_q}(Q_{cam})_{t_q}. \tag{2}$$

In this way, we *fix* the query point to the corresponding object, and then obtain its track across the whole video by leveraging the object's pose annotation. For any timestamp $t$, the position of the query point can be then obtained by

$$(Q_{cam})_t = (P_{cam}^w)_t(P_w^{obj})_t Q_{obj}. \tag{3}$$

To compute the visibility $v$ of the query point at any time, we first employ a pretrained semantic segmentation model to obtain the semantic masks of the operator's hands $\{H_t\}$, as these are not modelled in the digital replica. Then, we compute the visibility $v$ by verifying that the query point's depth is close to the observed depth, and it does not lie on the hands segmentation mask $H$, so

---

*We use the notation $P_b^a$ to represent the SE(3) transform from coordinate frame $a$ to frame $b$.

†We use the notation $Q_{cam}$ and $Q_{obj}$ to denote the position of the 3D point $Q$ in the camera and object coordinate frames, respectively.

$$v_t = \mathbb{1}(|(Z(Q_{cam}) - D_t(u,v)| < \delta) \cdot (1 - H_t(u,v)), \tag{4}$$

where $(u,v) = \Pi_K((Q_{cam})_t)$ is the projection of the query point $(Q_{cam})_t$ to the image plane according to the given camera intrinsics $K$, and $Z((x,y,z)) = z$ is the function that extracts the z-component of a 3D point. This approach allows us to compute the 3D trajectory $\{(Q_{cam})_t\}$ and visibility flag $\{v_t\}$ of the query point across the whole video.

### 3.2 DriveTrack

The DriveTrack split is based on videos from the Waymo Open dataset [50], and the 2D point trajectory pipeline used in DriveTrack [3]. In particular, each frame $I_t$ in a video sequence $V$ has a corresponding, time-synchronized point cloud $\{C_t\}$ from the Waymo car's LIDAR. The subset of points that correspond to a randomly selected, single, tracked vehicle $(Q_{cam})_{t_s}$ are subselected from the entire point cloud $(Q_{cam})_{t_s} \subset C_{t_s}$ at a certain sampling time $t_s$ using a manually-annotated 3D bounding box around the chosen object. These object-specific points are then tracked across the whole video using: (i) a vehicle rigidity assumption, and, (ii) the pose and position of the object's annotated 3D bounding box through the entire video sequence. Specifically, the tracked points in object coordinate frame $Q_{obj}$ are first computed using (2), and then the trajectories in camera coordinate frame $\{(Q_{cam})_t\}$ are obtained using (3).

Visibility flag is estimated by first computing the dense depth map $D_t$ of the corresponding camera video frames through interpolation of sparse LIDAR values as in [3]. This is compared to the point's depth computed from the 3D point trajectory given by $(Q_{cam})_t$, as in the first term of (4). If the point depth (distance from the camera center to the query point) is greater than the depth provided by the depth map (with a 5% relative threshold margin), it is marked as not visible.

Finally, to determine the 2D query points $q = (x_q, y_q, t_q)$ we sample $t_q$ uniformly among the visible timestamps $v_t$, and then obtain $(x_q, y_q) = \Pi_K((Q_{cam})_{t_q})$.

### 3.3 Panoptic Studio

The original Panoptic Studio dataset [21] consists in video sequences captured inside a recording dome using stationary cameras, and depicting different actors performing various actions such as passing a ball or swinging a tennis racket, featuring complex non-rigid motions. To obtain 3D trajectories, we leverage the pretrained dynamic 3D reconstructions from Luiten et al. [35]. These have been obtained by first performing a rigid 3D reconstruction through 3D Gaussian Splatting [23], fitting a set of 3D Gaussians $\{(\mu_i, \Sigma_i)_{t_0}\}_{i=1,...,N}$ to the first timestamp $t_0$ of each sequence using the multiple cameras available in the dome. Then, these Gaussians are displaced and rotated in a as-rigid-as-possible manner to model the motion occurring in the subsequent frames of the video. For more details, please refer to [35]. Using these pretrained splatting models, we render pseudo-ground-truth depth maps $\{D_t\}$ for each sequence. Then, given an query point $q = (x_q, y_q, t_q)$, we unproject the point to 3D following (1), obtaining $(Q_{cam})_{t_q}$, and retrieve the index $i^*$ of the closest Gaussian center at that time using the poses $\{(P_{cam}^w)_t\}$, so that $i^* = \underset{i=1,...,N}{\operatorname{argmin}} \|(Q_{cam})_{t_q} - (P_{cam}^w)_{t_q}(\mu_i)_{t_q}\|_2$.

Note that due to the distance between $(Q_{cam})_{t_q}$ and $(P_{cam}^w)_{t_q}(\mu_{i^*})_{t_q}$, their projections onto the image plane will not match exactly. To account for this difference, we adjust the query point's 2D position as $(x_q, y_q) = \Pi_K(P_{cam}^w)_{t_q}(\mu_{i^*})_{t_q}$. Then, the query point's 3D track come by following the motion of the $i^*$ Gaussian center, so $(Q_{cam})_t \equiv (P_{cam}^w)_t(\mu_{i^*})_t$.

Visibility $\{v_t\}$ is estimated as in (4) (omitting the second term), by comparing the depth of our 3D query point with the observed depth from $D_t$. We only track points across the foreground deforming characters, as tracking background points is trivial given that the cameras in this dataset are stationary.

### 3.4 Data Cleanup and Validation

While the aforementioned procedures generally produce high quality trajectories, small amounts of noise from the underlying dataset sources can cause issues in a small fraction of sequences. These minor inaccuracies, for example, can be caused by small misalignment between Aria Digital Twins synthetic annotations and the real world video, LIDAR sensor noise, insufficiently constrained Gaussian splats, or numerical error. We minimized these errors through automated methods, and then manually checked a sampling of the videos to ensure accuracy.

Firstly, since trajectories are descriptors of surface motion, their motion should be localized to their associated object. We use instance segmentation models to generate object masks for each frame [25], filtering out errant trajectories that exceed these boundaries when not occluded. In DriveTrack specifically, the tracked bounding box is an approximation to the true object mesh, but such errors are fixed with tight segmentation masks (trimming 2-3% of initial trajectories).

Secondly, we observed that some trajectories have a 'flickering' visibility flag. This is not unique to TAPVid-3D, as we notice this in the widely used Kubric [14] and DriveTrack [3]. However, to mitigate this in our dataset, we oversample trajectories in the annotation generation pipeline, and filter trajectories whose visibility state changes more times than 10% of the total number of video frames. More details can be found in the supplementary.

## 3.5 Metrics

To accompany the TAPVid-3D dataset, we adopt and extend the metrics used in TAP-2D [10] to the 3D tracking domain. These metrics measure the quality of the predicted 3D point trajectories (APD), the ability to predict point visibility (OA), or both simultaneously (AJ).

The **APD** ($< \delta_{avg}^{x}$) metric measures the average percent of points within $\delta$ error. If $\hat{P}_t^i$ is the $i$'th point prediction at time $t$, $P_t^i$ is the corresponding ground truth 3D point, and $v_t^i$ is the ground-truth point visibility flag, then:

$$\text{APD}_{3D} \equiv \frac{1}{V} \sum_{i,t} v_t^i \cdot \mathbb{1}(\|\hat{P}_t^i - P_t^i\| < \delta_{3D}(P_t^i)), \tag{5}$$

where $\mathbb{1}(\cdot)$ is the indicator function, $\|\cdot\|$ is the Euclidean norm, $V = \sum_{i,t} v_t^i$ is the total number of visibles, and $\delta_{3D}(P_t^i)$ is the threshold value.

The value of this threshold is *relative to ground-truth depth*; and is defined by *unprojecting* a pixel threshold $\delta_{2D} \subset \{1, 2, 4, 8, 16\}$ to 3D space using the camera intrinsic parameters: $\delta_{3D}(P_t^i) = Z(P_t^i) \cdot \delta_{2D}/f$, where $f$ is the camera focal length. We argue that points that are closer to the camera are of higher importance than those that are far away, which is enforced by the definition of $\delta_{3D}$[‡]. Note that by using this definition, the APD$_{3D}$ metric defined in (5) is numerically equivalent to the APD$_{2D}$ metric from [10] when the point estimations $\hat{P}_t^i$ all have correct depths.

In addition, it is important to distinguish between occluded and visible points, because downstream algorithms often want to rely exclusively on predictions which are based on visual evidence, rather than on the guesses that might be very wrong. To this end, we adopt the occlusion accuracy metric (**OA**) from [10], which computes the fraction of points on each trajectory where $\hat{v}_t^i = v_t^i$, where $\hat{v}_t^i$ is the model's (binary) visibility prediction.

Finally, we define **3D-AJ**, 3D Average Jaccard, following TAP, which combines OA and APD$_{3D}$. The AJ metric calculates the number of *true positives* (number of points within the $\delta_{3D}$ threshold, predicted correctly to be visible), divided by the sum of *true positives* and *false positives* (predicted visible, but are occluded or farther than the threshold) and *false negatives* (visible points, predicted occluded or predicted to exceed the threshold). Mathematically, it is defined as:

$$\text{AJ}_{3D} \equiv \frac{\sum_{i,t} v_t^i \, \hat{v}_t^i \, \alpha_t^i}{\sum_{i,t} v_t^i + \sum_{i,t}\left((1 - v_t^i)\, \hat{v}_t^i\right) + \sum_{i,t}\left(v_t^i \, \hat{v}_t^i \,(1 - \alpha_t^i)\right)}, \tag{6}$$

where $\alpha_t^i = \mathbb{1}(\|\hat{P}_t^i - P_t^i\| < \delta_{3D}(P_t^i))$ indicates whether the point prediction is below the distance threshold.

Note the relationship between this and prior metrics: if the depth estimates for a given point are perfect, then this metric reduces to 2D-AJ, as there will be no depth errors. It is also related to a common Monodepth's metric $\delta < 1.25^k$, which also places a hard threshold on the *relative* depth of the estimated point w.r.t. the ground truth; if the video is a single frame (occlusion-free) and the query points are dense, then there should be no 2D errors, and our metric will behave like a Monodepth metric. For general sequences, however, the algorithm must output correct tracking *and* correct depth—i.e., a full 3D trajectory—in order to be considered correct by our metric.

---

[‡]A similar depth-adaptive threshold approach is used in Monodepth literature [43].

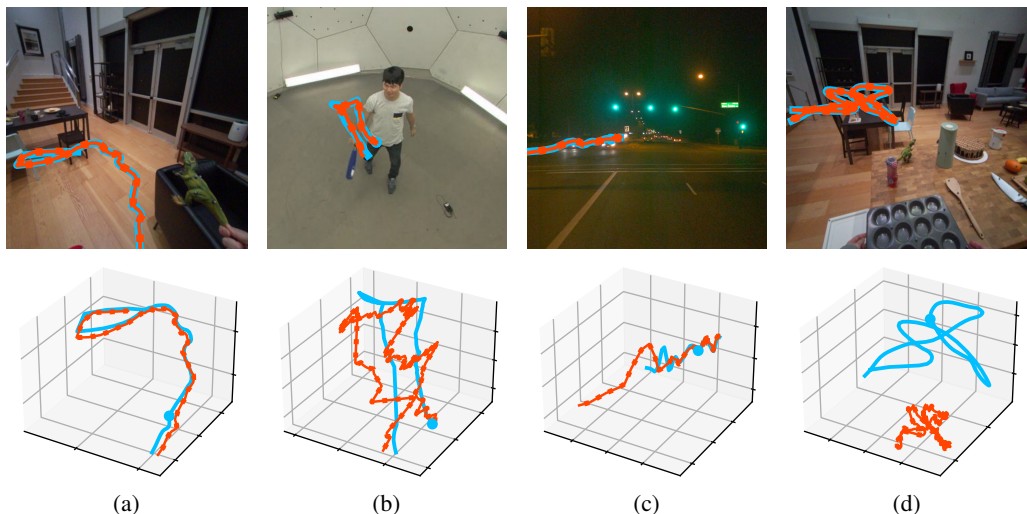

|  (a)  |  (b)  |  (c)  |  (d)  |

Figure 2: Illustrative TAP-3D results of BootsTAPIR+ZoeDepth. We compare the ground-truth 2D and 3D tracks (blue solid) with the predicted tracks (red dotted). (a) Accurate tracking. (b) Noisy depth estimations result in a noisy 3D track. (c) Inconsistent depth scales across time (scale drift). (d) Inconsistent depth scales across space don't allow a single global scale factor to properly fit all tracks.

Finally, one additional complication is *scale ambiguity*. As in monocular depth literature [43], we globally re-scale predictions to match ground truth, before computing the metrics. We do this by multiplying predictions $\hat{P}_t^i$ by the median of the depth ratios $\|P_t^i\|/\|\hat{P}_t^i\|$ over all points and frames, which we call *global median* rescaling. Note, however, that some algorithms may be more adept at estimating the relative depth of *individual* points: algorithms trained in simulation, for instance, may be able to estimate the depth change for a single point just by analyzing the frequencies. With a slight change to the rescaling, we can accomodate such methods even when they don't produce consistent scale between points. Therefore, we define the *per-trajectory* rescaling, which rescales each track $P^i$ separately multiplying by $\|P_{t_q}^i\|/\|\hat{P}_{t_q}^i\|$, where $t_q$ is the query timestamp. Finally, we explore a hybrid approach—which considers scaling in local neighborhoods to better account for object-object interactions—in supplementary.

## 4   Baselines on TAPVid-3D

We construct our main baselines by a combination of state-of-art 2D point trackers and monocular depth estimators. In particular, we use: (a) for 2D tracking, several state-of-the-art models such as CoTracker [22], BootsTAPIR model [12], and TAPIR [11]; and (b) for depth regression, both the monocular depth estimation model ZoeDepth [4] and the SfM pipeline COLMAP [46, 47]. To convert the frame pixel space predictions into metric $x, y$-position coordinates, we unproject using the camera intrinsics and the $z$-estimate provided by ZoeDepth. In the case of COLMAP, we import the 2D trajectories produced by the TAP methods before running the SfM reconstruction pipeline. Results for the corresponding baseline methods are shown in Table 4. We provide inference settings and details on used compute resources in the supplemental material.

In the lower half of Table 4, we provide results on the `minival` split, including the one very recently released work [60] that targets the nascent TAP-3D task. As SpatialTracker [60] was released three days before writing and submission of the current manuscript, we were only able to evaluate this method on `minival` for the moment. In pursuit of methods that jointly learn 2D and depth tracking, we additionally trained a 3D version of TAPIR (TAPIR-3D), trained *only* on the synthetic Kubric dataset [14]. TAPIR-3D predicts $x, y, z$ jointly per trajectory. Using an optimal per-trajectory depth scaling, TAPIR3D achieves an 3D-AJ of 9.4 AJ, slightly trailing our TAPIR + ZoeDepth baseline when using. More details are in the supplemental.

Comparing Tables 5 and 4, we find that the 3D tracking performance of our baselines are significantly lower compared to their effective 2D tracking abilities. We show examples in Figure 2, illustrating common failure modes regressing 3D trajectories, noting that while the 2D trajectories look accurate, their understanding of total scene geometry and correct 3D motion is poor.

| Baseline | Aria 3D-AJ ↑ | APD ↑ | OA ↑ | DriveTrack 3D-AJ ↑ | APD ↑ | OA ↑ | PStudio 3D-AJ ↑ | APD ↑ | OA ↑ | Average 3D-AJ ↑ | APD ↑ | OA ↑ |
|---|---|---|---|---|---|---|---|---|---|---|---|---|
| Static Baseline | 4.9 | 10.2 | 55.4 | 3.9 | 6.5 | 80.8 | 5.9 | 11.5 | 75.8 | 4.9 | 9.4 | 70.7 |
| TAPIR + COLMAP | 7.1 | 11.9 | 72.6 | 8.9 | 14.7 | 80.4 | 6.1 | 10.7 | 75.2 | 7.4 | 12.4 | 76.1 |
| CoTracker + COLMAP | 8.0 | 12.3 | 78.6 | 11.7 | **19.1** | 81.7 | 8.1 | 13.5 | 77.2 | **9.3** | 15.0 | 79.1 |
| BootsTAPIR + COLMAP | 9.1 | 14.5 | 78.6 | **11.8** | 18.6 | 83.8 | 6.9 | 11.6 | 81.8 | **9.3** | 14.9 | 81.4 |
| TAPIR + ZoeDepth | 9.0 | 14.3 | 79.7 | 5.2 | 8.8 | 81.6 | 10.7 | 18.2 | 78.7 | 8.3 | 13.8 | 80.0 |
| CoTracker + ZoeDepth | **10.0** | 15.9 | 87.8 | 5.0 | 9.1 | 82.6 | 11.2 | **19.4** | 80.0 | 8.7 | 14.8 | 83.4 |
| BootsTAPIR + ZoeDepth | 9.9 | **16.3** | 86.5 | 5.4 | 9.2 | **85.3** | 11.3 | 19.0 | **82.7** | 8.8 | 14.8 | 84.8 |
| BootsTAPIR + DepthAnythingV2 | 3.7 | 6.9 | 86.5 | 7.4 | 12.4 | 85.4 | 5.6 | 10.3 | **82.7** | 5.5 | 9.9 | **84.9** |
| CoTracker + DepthAnythingV2 | 3.6 | 6.6 | 87.8 | 7.1 | 12.6 | 82.6 | 5.6 | 10.5 | 80.0 | 5.4 | 9.9 | 83.5 |
| TAPIR + DepthAnythingV2 | 3.3 | 6.1 | 79.6 | 6.9 | 11.5 | 81.7 | 5.3 | 9.8 | 78.7 | 5.1 | 9.1 | 80.0 |
| TAPIR-3D | 2.5 | 4.8 | 86.0 | 3.2 | 5.9 | 83.3 | 3.6 | 7.0 | 78.9 | 3.1 | 5.9 | 82.8 |
| SpatialTracker [60] | 9.9 | 16.1 | **89.0** | 6.2 | 11.1 | 83.7 | 10.9 | 19.2 | 78.6 | 9.0 | **15.5** | 83.7 |
| BootsTAPIR + COLMAP* | 7.3 | 11.5 | 76.3 | **9.3** | 15.1 | 83.5 | 6.2 | 10.6 | 78.7 | 7.6 | 12.4 | 79.5 |
| BootsTAPIR + ZoeDepth* | 8.6 | 14.5 | 86.9 | 5.1 | 8.7 | **83.5** | 10.2 | 17.7 | 82.0 | 8.0 | 13.6 | **84.1** |
| SpatialTracker [60]* | **9.2** | 15.1 | 89.9 | 5.8 | 10.2 | 82.0 | 9.8 | **17.7** | 78.4 | **8.3** | 14.3 | 83.4 |

Table 4: We compare the performance of several 2D-TAP models [11, 12, 22] combined with ZoeDepth [4], Depth Anything V2 [61], and COLMAP [46] on the TAPVid-3D benchmark. We report the proposed 3D-AJ as well as the APD and OA metrics. Rows marked with * indicate evaluation on the `minival` split. Occlusion Accuracy (OA) measures accuracy of point visibility classification, Average Position Error within Delta measures the point position error, and the headline metric, 3D Average Jaccard (3D-AJ), combines these two as a measure of overall performance on the TAP-3D task.

| | Aria 2D-AJ ↑ | DriveTrack 2D-AJ ↑ | PStudio 2D-AJ ↑ | Total 2D-AJ ↑ | APD ↑ | OA ↑ |
|---|---|---|---|---|---|---|
| TAPIR [11] | 48.6 | 57.2 | 48.7 | 53.2 | 67.4 | 80.5 |
| CoTracker [22] | 54.2 | 59.8 | 51.0 | 57.2 | 74.2 | 84.5 |
| BootsTAPIR [12] | **54.7** | **62.9** | **52.4** | **59.1** | **74.7** | **85.6** |

Table 5: Evaluating the 2D point tracking performance of our baseline models on TAPVid-3D data, by projecting the ground truth 3D trajectories onto the 2D frame.

**Limitations and Responsible Usage.** The three data sources in our benchmark do not cover all possible domains where users may want to infer dynamics, and automatic annotations may be imperfect, e.g. where the underlying sensor readings have noise (despite our filtering). Furthermore, we inherit some limitations from TAP and monocular depth: we only evaluate tracking for solid, opaque objects. From an ethical perspective, Aria and Panoptic were collected in lab settings with consenting participants, while DriveTrack's Waymo videos come from public roads in 6 US cities. This paper may inherit biases from these datasets, e.g., participants are lab researchers or the populations of those 6 US cities. This dataset is not intended for training, but care should be taken in training data to avoid biases. As a benchmark, the broader impacts are similar to those in prior vision and tracking works: there may be very long term applications to activity recognition and surveillance.

## 5 Conclusion

We introduce TAPVid-3D, a new benchmark for evaluating the nascent Tracking Any Point in 3D (TAP-3D) task. We contributed (1) a pipeline to annotate three distinct real-world video datasets to produce 3D correspondence annotations per video, (2) the first metrics for the TAP-3D task, which measure multiple axis of accuracy of 3D track estimates, and (3) an assessment of the current state of TAP-3D, by evaluating commonly used tracking models. We believe this benchmark will accelerate research on the TAP-3D task, allowing the development of models with greater dynamic scene understanding from monocular video.

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
