# Supplemental Material for TAPVid-3D

## Table of Contents

## 1. More Dataset Samples

We provide visualizations illustrating each of the TAPVid-3D dataset splits in HTML in the following files: `pstudio.html`, `adt.html`, `drivetrack.html`, which display the mp4 files also available in the `video_visualization` folder, as well as links to interactive 3D visualizations. We included as many samples as we could fit within the 50MB supplementary size limit.

In addition, we provide the reviewers with a Colab Notebook*, which enables interested reviewers to read, load, interact with, and generate visualizations for all the data in the `minival` split (containing samples from all three constituent data sources). This can help with understanding the format and contents of each dataset example. Reviewers can run this Colab notebook online at:

https://colab.research.google.com/drive/1Ro2sEOlAvq-hOlixrUBBOoTYXEwXNr66

Finally, we include static visualizations of trajectories in the figures included in the Visualized Samples section at the end of this PDF.

## 2. Dataset Statistics

Figure 1 showcases various summary statistics about the TAPVid-3D datasets and its 3D point trajectories. In the top left, we have the distribution of the number of frames in each video. The ADT-sourced videos contain the longest videos, and clips of 300 frames were extracted. Similarly Panoptic Studio contains clips of 150 frames, while DriveTrack contains clips of varying duration. In the top right, we have the number of point tracks annotated in each clip. In the bottom right, we count the number of 'static' trajectories in each video, marking a trajectory as static if the distance between all pairwise locations within a single point's trajectory is less than 1 centimeter. The roughly 10 DriveTrack videos consisting of static trajectories are usually cars stopped at stoplights. These 'static' videos are a small minority of the 4000+ clips in TAPVid-3D. In the bottom right, we show the average velocity of each trajectory in the dataset, noting that trajectories in DriveTrack are the fastest. These histograms convey that there is a diversity of overall trajectory lengths, video lengths, and point velocities in the TAPVid-3D dataset. Additionally, this dataset is larger than two widely used 2D point tracking real-world-video datasets: TAPVid-Kinetics (1,189 videos) and TAPVid-DAVIS (30 videos).

## 3. Metrics using *Median*, *Per-Trajectory*, and *Local Neighborhood* Rescaling

In the results included in the main paper, we compute the 3D Average Jaccard and APD metrics using a *global median* rescaling procedure (L277). To get a good score, the entire scene must be

---

*The Colab does not collect any view analytics, or track visitors (in any way accessible to the authors of this work).

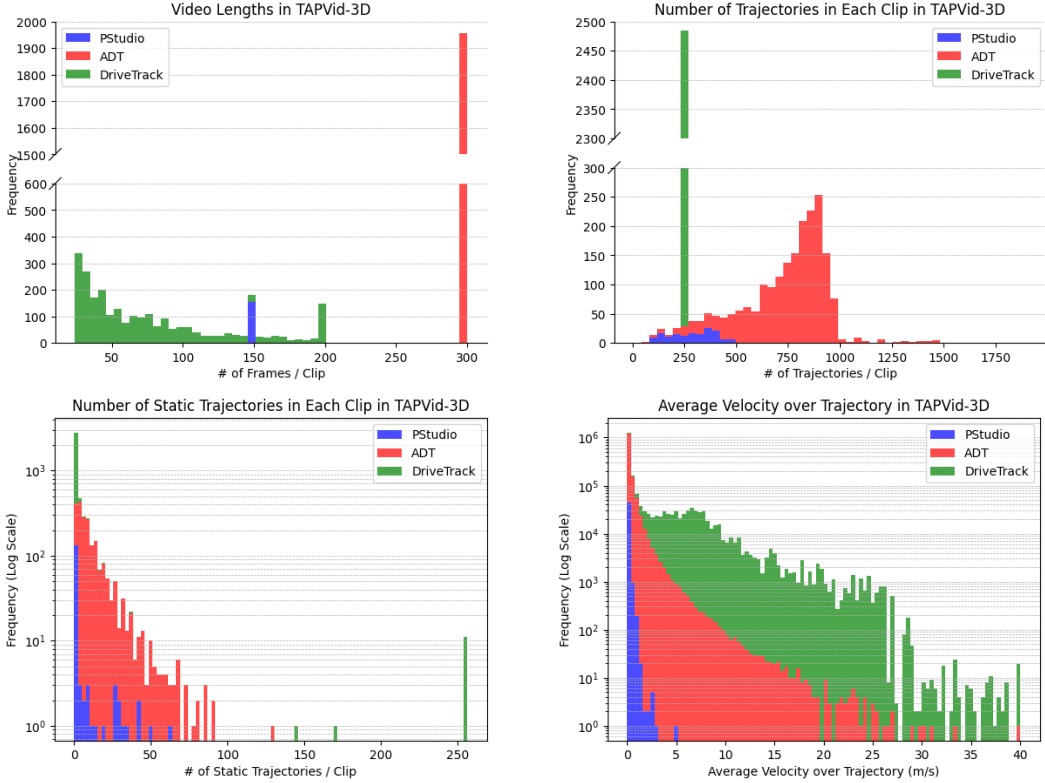

Figure 1: Statistics on TAPVid-3D. Top left: video lengths. Top right: Number of trajectories in each clip. Bottom left: Number of static tracks in each clip. Bottom right: average point velocity.

reconstructed up to scale, and dynamic objects must be placed precisely. This is useful for many applications, such as navigation, but for others it may be overly stringent. If there is little camera motion, or if some of the objects have unclear size, it may also be very difficult for models to infer global scene shape.

However, not all applications require such strong global scene shape capabilities. For example, for imitation learning, we may want an agent to simply approach an object. For such applications, measuring the relative depth of estimated 3D locations along a *single trajectory* may be sufficient, and it may be substantially easier, as the (2D) scaling of local textures may provide enough information to solve the problem. More generally, for robotic imitation of an assembly task, it is the *local* consistency that's most important: as long as points that are near each other in 3D have the correct depth relative to one another, then the relative pose of the assembled parts will be clear, especially at the critical stage when the assembled parts are close together.

To enable more rapid progress in such domains, we propose two additional approaches to rescaling estimated trajectories to match the ground-truth 3D point cloud: *Per-Trajectory*, and *Local Neighborhood*. We apply the same Average Jaccard metric regardless of how the points are rescaled, although in the case of Local Neighborhood, the ground truth trajectories are also slightly modified, as explained below. In all cases, users can evaluate the same predictions using any metric without providing any extra information.

Per-Trajectory scaling is computed by rescaling each track $P^i$ separately, multiplying by $\|P^i_{t_q}\|/\|\hat{P}^i_{t_q}\|$, where $t_q$ is the query frame index, and then computing 3D AJ as before. As a result, methods must only compute the *relative* depth for the point at each time, relative to the query frame.

Local Neighborhood is somewhat more involved. Here, the goal is to capture whether *nearby* points are scaled correctly relative to one another, even if distant parts of the scene may not be. For example, if the goal is to understand an action depicted in a video that uses a tool, it is typically important to understand where the hand is relative to the tool, and where the tool is relative to the objects it's acting on. The distance to the backgrounds–such as the back wall of the room–may not be obvious, especially if there's relatively little camera motion. However, precisely computing these distances is not relevant to understanding the tool's motion.

Intuitively, we wish to find an intermediate between two extremes: either rescaling the entire scene with a single scale factor, or rescaling every point with its own scale factor. To this end, we propose to scale each track according to the other track segments that intersect with its *4D tubelet* [8], according to a fixed neighborhood radius.

Specifically, we start by choosing a single neighborhood radius $\tau$ specified in meters. For a given trajectory $P^i$, we first find all points that are within $\tau$ meters of the ground truth on any frame, which define the tubelet $\mathcal{T}(P^i)$ associated to the trajectory $P^i$:

$$\mathcal{T}(P^i) = \{P_t^j \ \ s.t. \ \|P_t^j - P_t^i\| < \tau\}.$$

Note that tubelet $\mathcal{T}(P^i)$ includes the trajectory $P^i$ entirely, plus the portions of the trajectories of the other points where they come closer than the tublet's radius $\tau$. For each selected ground-truth point in the tubelet, we select the corresponding points from the predictions to construct $\mathcal{T}(P^i; \hat{P}^i)$ as

$$\mathcal{T}(P^i; \hat{P}^i) = \{\hat{P}_t^j \ \ s.t. \ \|P_t^j - P_t^i\| < \tau\}.$$

Analogously, we select the predicted and ground-truth visibility $\mathcal{T}(\hat{v}^i)$ and $\mathcal{T}(v^i)$.

Finally, given a predicted tubelet $\mathcal{T}(P^i; \hat{P}^i)$, we rescale all its points together using the ratio of query point distances $\|P_{t_q}^i\|/\|\hat{P}_{t_q}^i\|$, and evaluate the rescaled tubelet set as if it were a single trajectory, by replacing $P^i$, $\hat{P}^i$ and $v_i$ with their tubelet counterparts in equations (5) or (6) to compute the $APD_{3D}$ and $AJ_{3D}$ respectively. In our experiments, we set the radius $\tau$ to 3 centimeters for the PStudio and Aria scenes, and as 10 centimeters for the DriveTrack scenes, across all experiments. This is because tabletop manipulation and human-object motion likely require finer-grained movement than large vehicle movement in the Waymo Open public road scenes.

| | Aria | | | DriveTrack | | | PStudio | | | **Average** | | |
|---|---|---|---|---|---|---|---|---|---|---|---|---|
| Baseline | 3D-AJ↑ | APD↑ | OA↑ | 3D-AJ↑ | APD↑ | OA↑ | 3D-AJ↑ | APD↑ | OA↑ | 3D-AJ↑ | APD↑ | OA↑ |
| BootsTAPIR + Zoedepth | 17.3 | 27.0 | 86.5 | 7.4 | 12.3 | 85.3 | 12.3 | 20.6 | 82.7 | 12.3 | 20.0 | 84.8 |
| CoTracker + Zoedepth | 17.4 | 26.3 | 87.8 | 6.7 | 12.3 | 82.6 | 12.0 | 20.8 | 80.0 | 12.0 | 19.8 | 83.4 |
| TAPIR + Zoedepth | 16.2 | 24.2 | 79.7 | 7.4 | 12.2 | 81.6 | 12.0 | 20.0 | 78.7 | 11.9 | 18.8 | 80.0 |
| BootsTAPIR + COLMAP | 28.8 | 41.3 | 78.6 | 20.0 | 29.3 | 83.8 | 12.9 | 20.8 | 81.8 | 20.6 | 30.4 | 81.4 |
| CoTracker + COLMAP | 26.8 | 38.3 | 78.6 | 18.2 | 28.8 | 81.7 | 12.1 | 19.7 | 77.2 | 19.1 | 28.9 | 79.1 |
| TAPIR + COLMAP | 26.5 | 37.7 | 72.6 | 16.5 | 24.6 | 80.4 | 12.1 | 19.6 | 75.2 | 18.4 | 27.3 | 76.1 |
| TAPIR-3D | 8.5 | 14.9 | 86.0 | 10.2 | 17.0 | 83.3 | 7.2 | 13.1 | 78.9 | 8.6 | 15.0 | 82.8 |
| SpatialTracker | 17.4 | 26.9 | 89.0 | 9.0 | 16.1 | 83.7 | 14.2 | 24.6 | 78.6 | 13.6 | 22.5 | 83.7 |
| Static Tracks | 5.4 | 11.8 | 55.4 | 4.8 | 8.4 | 80.8 | 6.4 | 12.7 | 75.8 | 5.5 | 11.0 | 70.7 |

Table 1: **Using per-trajectory depth scaling**. We compare the performance on the `full_eval` split of several 2D-TAP models [3, 4, 6] combined with ZoeDepth [2] and COLMAP [10] on the TAPVid-3D benchmark. We also measure performance on the recently released SpatialTracker [12], and a static point baseline, in which the predicted trajectories are exactly the same as the query point.

## 4. Evaluations with Median, Per-Trajectory, and Local Neighborhood Scaling

Tables 1, 2, 3 present additional experimental results on the `full_eval` set, on all our baselines. To avoid biasing the results to the TAPVid3d splits with higher number of videos, these tables present averaged results across the three constituent data sources (weighing each source equally, rather than weighted by dataset size, as was done in Table 3 in the main paper).

| Baseline | Aria | | | DriveTrack | | | PStudio | | | Average | | |
|---|---|---|---|---|---|---|---|---|---|---|---|---|
| | 3D-AJ ↑ | APD ↑ | OA ↑ | 3D-AJ ↑ | APD ↑ | OA ↑ | 3D-AJ ↑ | APD ↑ | OA ↑ | 3D-AJ ↑ | APD ↑ | OA ↑ |
| BootsTAPIR + Zoedepth | 16.8 | 26.3 | 86.7 | 6.4 | 10.9 | 85.3 | 11.6 | 19.6 | 82.6 | 11.6 | 18.9 | 84.9 |
| CoTracker + Zoedepth | 17.0 | 25.7 | 88.0 | 6.0 | 10.9 | 82.6 | 11.4 | 19.9 | 80.0 | 11.4 | 18.8 | 83.5 |
| TAPIR + Zoedepth | 15.7 | 23.5 | 79.8 | 6.3 | 10.5 | 81.6 | 11.2 | 18.9 | 78.7 | 11.0 | 17.6 | 80.1 |
| BootsTAPIR + COLMAP | 26.1 | 38.0 | 78.8 | 16.6 | 25.1 | 83.8 | 10.8 | 17.8 | 81.8 | 17.8 | 27.0 | 81.5 |
| CoTracker + COLMAP | 24.6 | 35.3 | 78.8 | 15.7 | 25.2 | 81.7 | 10.8 | 17.7 | 77.1 | 17.0 | 26.1 | 79.2 |
| TAPIR + COLMAP | 23.8 | 34.4 | 72.8 | 12.9 | 20.3 | 80.4 | 9.9 | 16.5 | 75.1 | 15.5 | 23.7 | 76.1 |
| TAPIR-3D | 7.3 | 12.9 | 86.3 | 5.9 | 10.5 | 83.4 | 5.1 | 9.6 | 78.9 | 6.1 | 11.0 | 82.8 |
| SpatialTracker | 16.7 | 25.7 | 89.3 | 6.9 | 12.4 | 83.7 | 12.3 | 21.6 | 78.5 | 12.0 | 19.9 | 83.8 |
| Static Tracks | 5.5 | 11.8 | 56.0 | 4.8 | 8.4 | 80.8 | 6.4 | 12.6 | 75.7 | 5.5 | 10.9 | 70.8 |

Table 2: **Using local neighborhood scaling**. We compare the performance on the `full_eval` split of 2D-TAP models [3, 4, 6] combined with ZoeDepth [2] and COLMAP [10]. We also include SpatialTracker [12], and a static point baseline.

| Baseline | Aria | | | DriveTrack | | | PStudio | | | Average | | |
|---|---|---|---|---|---|---|---|---|---|---|---|---|
| | 3D-AJ ↑ | APD ↑ | OA ↑ | 3D-AJ ↑ | APD ↑ | OA ↑ | 3D-AJ ↑ | APD ↑ | OA ↑ | 3D-AJ ↑ | APD ↑ | OA ↑ |
| BootsTAPIR + Zoedepth | 9.9 | 16.3 | 86.5 | 5.4 | 9.2 | 85.3 | 11.3 | 19.0 | 82.7 | 8.8 | 14.8 | 84.8 |
| CoTracker + Zoedepth | 10.0 | 15.9 | 87.8 | 5.0 | 9.1 | 82.6 | 11.2 | 19.4 | 80.0 | 8.7 | 14.8 | 83.4 |
| TAPIR + Zoedepth | 9.0 | 14.3 | 79.7 | 5.2 | 8.8 | 81.6 | 10.7 | 18.2 | 78.7 | 8.3 | 13.8 | 80.0 |
| BootsTAPIR + COLMAP | 9.1 | 14.5 | 78.6 | 11.8 | 18.6 | 83.8 | 6.9 | 11.6 | 81.8 | 9.3 | 14.9 | 81.4 |
| CoTracker + COLMAP | 8.0 | 12.3 | 78.6 | 11.7 | 19.1 | 81.7 | 8.1 | 13.5 | 77.2 | 9.3 | 15.0 | 79.1 |
| TAPIR + COLMAP | 7.1 | 11.9 | 72.6 | 8.9 | 14.7 | 80.4 | 6.1 | 10.7 | 75.2 | 7.4 | 12.4 | 76.1 |
| TAPIR-3D | 2.5 | 4.8 | 86.0 | 3.2 | 5.9 | 83.3 | 3.6 | 7.0 | 78.9 | 3.1 | 5.9 | 82.8 |
| SpatialTracker | 9.9 | 16.1 | 89.0 | 6.2 | 11.1 | 83.7 | 10.9 | 19.2 | 78.6 | 9.0 | 15.5 | 83.7 |
| Static Tracks | 4.9 | 10.2 | 55.4 | 3.9 | 6.5 | 80.8 | 5.9 | 11.5 | 75.8 | 4.9 | 9.4 | 70.7 |

Table 3: **Using median depth scaling**. We compare the performance on the `full_eval` split of 2D-TAP models [3, 4, 6] combined with ZoeDepth [2] and COLMAP [10]. We also include SpatialTracker [12], and a static point baseline.

As expected, the AJ increases when using the less-strict local rescaling approaches. That is, *per-trajectory* scaling require less scale consistency than the *local neighborhood* metric, which itself is less stringent than the *global median scaling*. However, different methods improve by different amounts. Perhaps most surprisingly, COLMAP gives strong performance with local and per-trajectory rescaling, but underperforms Zoedepth on Aria and Panoptic Studio when evaluated with global rescaling. This is likely because COLMAP completely fails to reconstruct moving content. For scenes where the majority of tracks are moving, the median rescaling will fail completely; therefore, Zoedepth giving reasonable estimates for a larger fraction of points gives it an advantage.

TAPIR-3D, unsurprisingly, presents poor performance using global or local scaling, as it does not provide relative depth estimates for different tracks. However, evaluated with per-trajectory scaling, it gives competitive results on DriveTrack, even outperforming SpaTracker. This is somewhat surprising given that it operates on a completely different principle than other methods; it is trained using entirely synthetic data and does not use any geometric constraints, nor relies on monodepth models providing geometric priors. Overall, the different strengths and weaknesses of these highly-diverse methods suggests that the best performance will come from a method that combines ideas from all three. SpaTracker is a step in this direction, using a monodepth initialization while checking the 2D consistency of 3D reconstructions via reprojection, similar to COLMAP, and it often gives competitive performance. We hope that this benchmark can provide a way to quantify how well future methods in this vein accomplish the task.

Finally, we include a static baseline, which predicts the static 3D point $P_q = K^{-1}[x_q, y_q, 1]^T \cdot Z_q$ for all timestamps, in order to quantify the impact of motion. Note that this baseline still requires ground truth depth for the query points, and so isn't trivial to reproduce automatically; however, it still performs very poorly, even for the PStudio dataset where the camera is static (note that the static baseline is static *in the camera coordinate frame*). Thus, we conclude tracking the camera and tracking the objects is important for obtaining strong performance.

## 5. Evaluations using Fixed Metric Distance Thresholds

In the Average Jaccard formulation in Section 3.5, we describe how we use determine correctly predicted points along a trajectory, using a depth-adaptive radius threshold denoted $\delta_{3D}(P_t^i)$. We also explored using a fixed metric threshold. Specifically, instead of the $\{1, 2, 4, 8, 16\}$ pixel thresholds (projected into 3D), we use a the fixed metric radius thresholds of 1 centimeter, 4 centimeter, 16 centimeters, 64 centimeters, and 2.56 meters. If the predicted point is within this distance to the ground truth point, it is marked as position correct within that threshold. Table 4 describes the model baselines results using this alternative metric.

| Baseline | Aria 3D-AJ↑ | APD↑ | OA↑ | DriveTrack 3D-AJ↑ | APD↑ | OA↑ | PStudio 3D-AJ↑ | APD↑ | OA↑ | Average 3D-AJ↑ | APD↑ | OA↑ |
|---|---|---|---|---|---|---|---|---|---|---|---|---|
| BootsTAPIR + Zoedepth | 31.9 | 45.6 | 86.5 | 11.4 | 16.3 | 85.3 | 41.5 | 55.9 | 82.7 | 28.3 | 39.2 | 84.8 |
| CoTracker + Zoedepth | 32.7 | 44.0 | 87.8 | 10.7 | 16.2 | 82.6 | 40.2 | 56.1 | 80.0 | 27.8 | 38.8 | 83.4 |
| TAPIR + Zoedepth | 28.2 | 41.7 | 79.7 | 11.0 | 15.9 | 81.6 | 39.0 | 55.2 | 78.7 | 26.1 | 37.6 | 80.0 |
| BootsTAPIR + COLMAP | 24.0 | 35.5 | 78.6 | 18.7 | 25.2 | 83.8 | 31.6 | 43.6 | 81.8 | 24.7 | 34.7 | 81.4 |
| CoTracker + COLMAP | 23.6 | 33.6 | 78.6 | 18.2 | 25.7 | 81.7 | 31.6 | 44.1 | 77.2 | 24.4 | 34.4 | 79.1 |
| TAPIR + COLMAP | 20.5 | 32.2 | 72.6 | 15.5 | 21.9 | 80.4 | 28.8 | 41.7 | 75.2 | 21.6 | 31.9 | 76.1 |
| TAPIR-3D | 19.2 | 29.9 | 86.0 | 7.0 | 10.9 | 83.3 | 24.8 | 36.1 | 78.9 | 17.0 | 25.6 | 82.8 |
| SpatialTracker | 33.1 | 45.0 | 89.0 | 13.1 | 19.3 | 83.7 | 39.5 | 56.1 | 78.6 | 28.5 | 40.2 | 83.7 |
| Static Tracks | 21.2 | 40.3 | 55.4 | 5.8 | 9.7 | 80.8 | 31.7 | 46.4 | 75.8 | 19.6 | 32.1 | 70.7 |

Table 4: **Using fixed metric thresholds for AJ, with median scaling**. We compare the performance on the `full_eval` split of several 2D-TAP models [3, 4, 6] combined with ZoeDepth [2] and COLMAP [10]. We also include SpatialTracker [12] and the static point baseline.

## 6. Baselines Details and Compute Resources

**CoTracker**. We use the pretrained model and PyTorch code from the official CoTracker codebase and run inference enabling the bi-directional tracking mode, with no other modifications to the default parameters. Internally, inference is performed in $512 \times 384$ resolution, and the output predictions are rescaled back to the original clip resolutions. Inference is performed using A100 GPUs, and processing each dataset clip takes about 30s, totaling roughly 38 GPUh for running CoTracker on the whole benchmark.

**BootsTAPIR and TAPIR**. We use the pretrained models and JAX code from the official codebase and run inference with the default parameters. Internally, inference is performed in $256 \times 256$ resolution, and the output predictions are rescaled back to the original clip resolutions. Inference was performed in a CPU cluster using up to 1024 CPUs and totalling about 22800 CPUh.

**COLMAP**. For running COLMAP, we dumped the 2D tracks estimated by CoTracker, TAPIR and BootsTAPIR as a set of per-frame image feature files and corresponding matches in txt format and imported those in COLMAP using the 'feature_importer' and 'matches_importer' functionality. We then perform 3D reconstruction through the incremental mapping pipeline ('mapper'). As each input 2D track can lead to multiple reconstructed 3D points across time (eg. for moving objects), we only keep those with larger "track length" (number of images where that 3D point was reconstructed from). Finally, we transform the resulting reconstructed 3D points positions in world coordinates to camera coordinates using the predicted extrinsic parameters. Inference was performed in a CPU cluster using up to 1024 CPUs and totalling about 14000 CPUh.

**ZoeDepth**. We used the pretrained models and PyTorch code from the official codebase and run inference with the default parameters. Inference is performed in the native resolution for the ADT and Panoptic Studio clips and in $720 \times 480$ for DriveTrack, where the original resolution was too large for running infrence in this model on a standard GPU. Inference was performed on 16 V100 GPUs, totalling about 200GPUh.

**TAPIR3D**

We propose a straightforward extension of TAPIR to 3D by training on 3D ground truth from Kubric [5]. As Kubric is synthetic data, it is straightforward to obtain ground-truth 3D point tracks. However, we don't expect that Monodepth models trained on Kubric will generalize, as the scenes are very different from real ones. However, we expect that a model trained here might be able to estimate *relative* depth of a point at different times on the trajectory, relative to the query point. Like with point tracking, we expect that low-level texture information may be sufficient to predict the relative depth (specifically, the scaling of the texture elements), and so high-level semantic understanding won't be necessary, meaning that it can be learned from a semantically meaningless dataset.

Specifically, we train TAPIR to output the log depth of each point on the trajectory, relative to the query point. This is a scalar quantity, and can be predicted using the same network structure as the other scalar quantites, e.g., the occlusion logit. That is, we predict an initial log-space scale factor for every frame by adding an extra head to TAPIR's occlusion prediction network, which performs convolutions on top of the cost volume for each frame followed by global average pooling. Then we feed this estimate to the iterative refinement steps by concatenating it with the local score maps, the initial occlusion estimate, and so on, passing it through the 1-D convolutional network which produces an update; again, we add an extra head on this network which produces and updated relative depth estimate. We apply an L1 loss on the estimate for both the initialization and each of the four refinement passes, with a weight of 1.0. Otherwise, we train the entire network using the procedure described in [3].

## 7.  Filtering Incorrect Trajectories

We apply different automatic filters for removing problematic tracks. Tracks can present three type of issues: (i) issues with visibility flags, (ii) queries which are outside the moving objects, and (iii) noisy 3D trajectories.

We found visibility issues (i) to be present in all dataset splits, and we remove it simply by oversampling the number of query points and discarding those whose visibility flag changes state more than a 10% of the number of frames in the video.

Issue (ii) was present mostly in the DriveTrack split, where trajectories in a video are localized and describe the motion of exactly one moving object in the scene. In some cases the 3D point-clouds associated with vehicles also contain points that are within the object bounding box, but outside of the object itself, such as in the road. To filter out errant trajectories, we use the Segment Anything model (SAM) to generate an object mask for each frame [7]. We prompt SAM with a point prompt, computed by taking the geometric median of DriveTrack trajectories at each point in time.

Finally, we found that noisy 3D trajectories (iii) could occur in the Panoptic Studio split, where sometimes the reconstructed 3D Gaussians where not sufficiently constrained due to surfaces having uniform colors. In this case we apply a similar approach as before, and score trajectories based on the percentage of time they are on foreground object masks across all camera viewpoints. We perform a hyperparameter search on the threshold value and select the points that stay on the object masks at least 75% of the time across all masks, which removes most of the problematic points.

## 8.  Dataset Specifications, Metadata, and other Details

For this dataset release, we preserve the licenses for the constituent original data sources, which are non-commercial. For our additions, to the extent that we can, we release under a standard Apache 2.0 license. A full amalgamated license will be available in the open-sourced repository during complete release of the work, after the review period is finished.

We will publicly host the dataset for wide consumption by researchers on Google Cloud Storage indefinitely. Part of the dataset is already hosted in this way (and how the Colab link linked above

is able to run). We also intend to open-source code for computing the new 3D-AJ metrics after the camera ready. We anticipate the release will require little maintenance (and the TAPVid-2D dataset release that the team released two years ago is similarly low maintanence), but we are happy to address any emergent issues raised by users.

Specific implementation details on how the dataset can be read are found in the Colab link provided. Each dataset example is provided in a `*.npy` file, containing the fields: `tracks_xyz` of shape [T, Q, 3] (containing the Q ground truth point tracks for the corresponding video of T frames, with $(x, y, z)$-coordinates in meters), `query_xyt` of shape [Q, 3] (containing each track's query point position in format (x, y, t), in (x,y)-pixel space and $t$ as the query frame index), the ground truth `visibility` flags with shape [Q, T], and the `camera_intrinsics` (as $[f_x, f_y, c_x, c_y]$). Each `*.npy` file is named after its corresponding video in the original data source, which can be loaded by downloading from the original hosting sites [1, 9, 11], respecting their corresponding licenses.

## 9.  Visualized Samples

See Figures 2, 3, 4, 5, 6, 7, 8, 9, and 10 below.

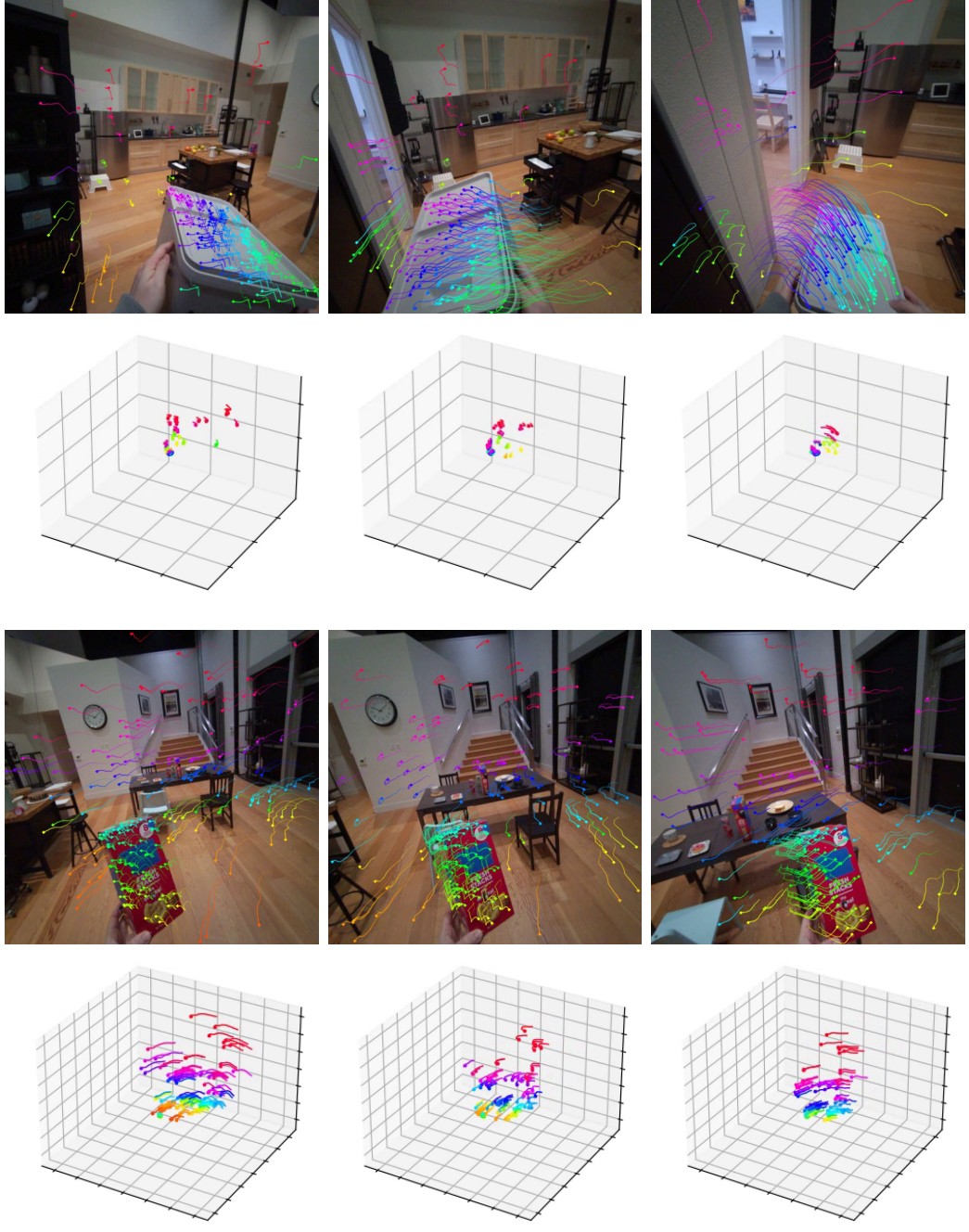

Figure 2: Random samples from ADT subset in TAPVid-3D: on the top row, we visualize the point trajectories projected into the 2D video frame; on the bottom row, we visualize the metric 3D point trajectories. For each video, we show 3 frames sampled at time step 30, 60 and 90.

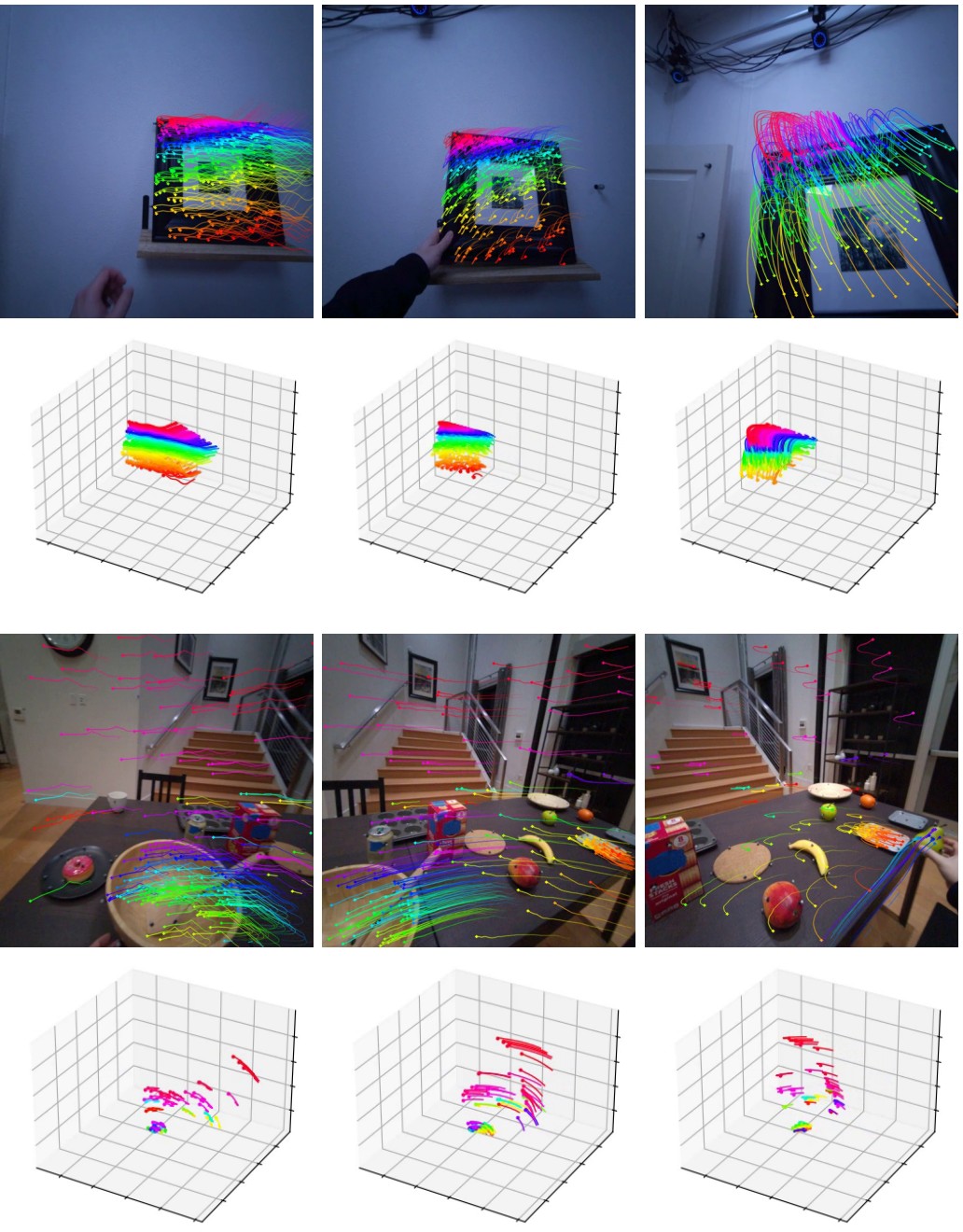

Figure 3: Random samples from ADT subset in TAPVid-3D (cont'd.): on the top row, we visualize the point trajectories projected into the 2D video frame; on the bottom row, we visualize the metric 3D point trajectories. For each video, we show 3 frames sampled at time step 30, 60 and 90.

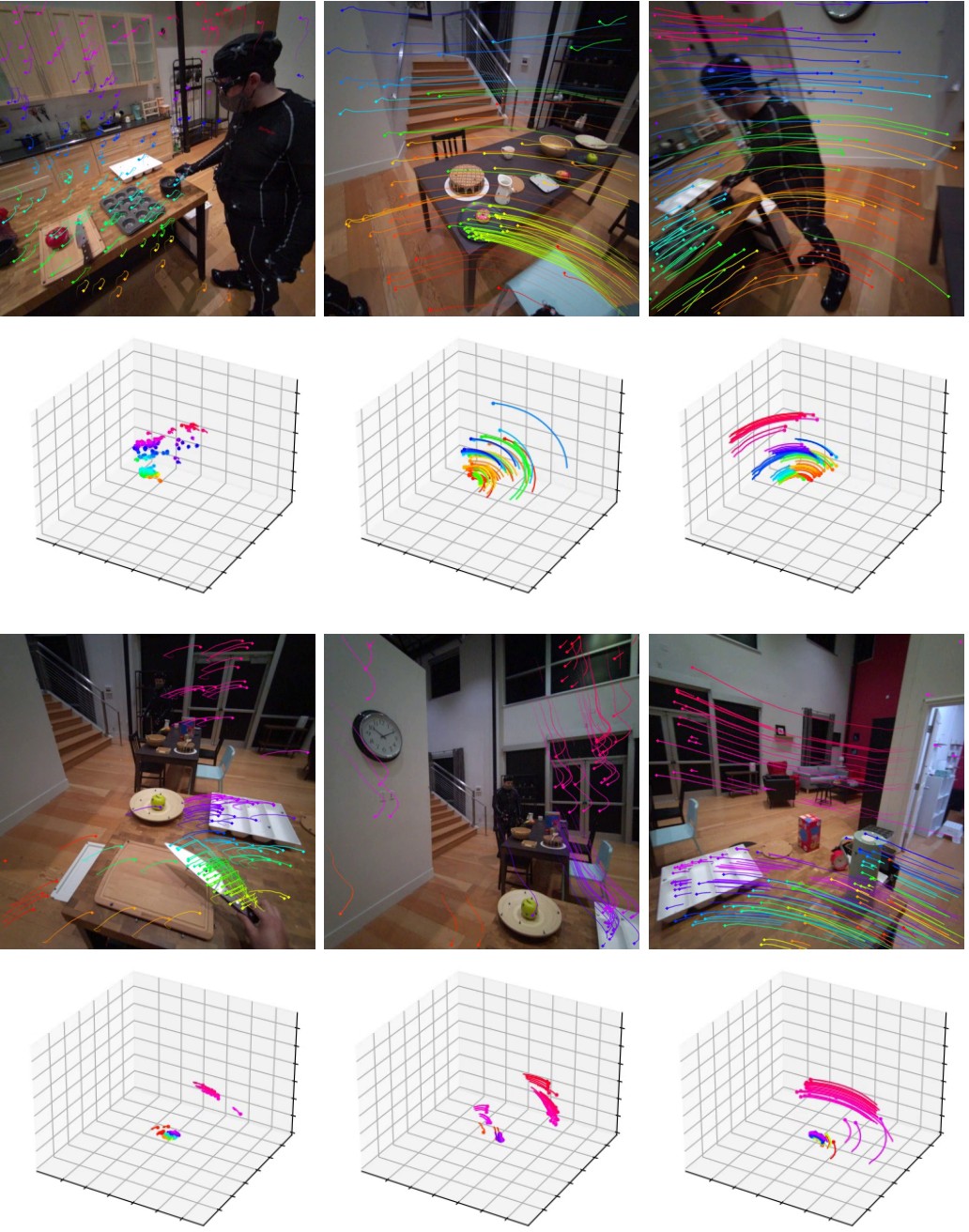

Figure 4: Random samples from ADT subset in TAPVid-3D (cont'd.): on the top row, we visualize the point trajectories projected into the 2D video frame; on the bottom row, we visualize the metric 3D point trajectories. For each video, we show 3 frames sampled at time step 30, 60 and 90.

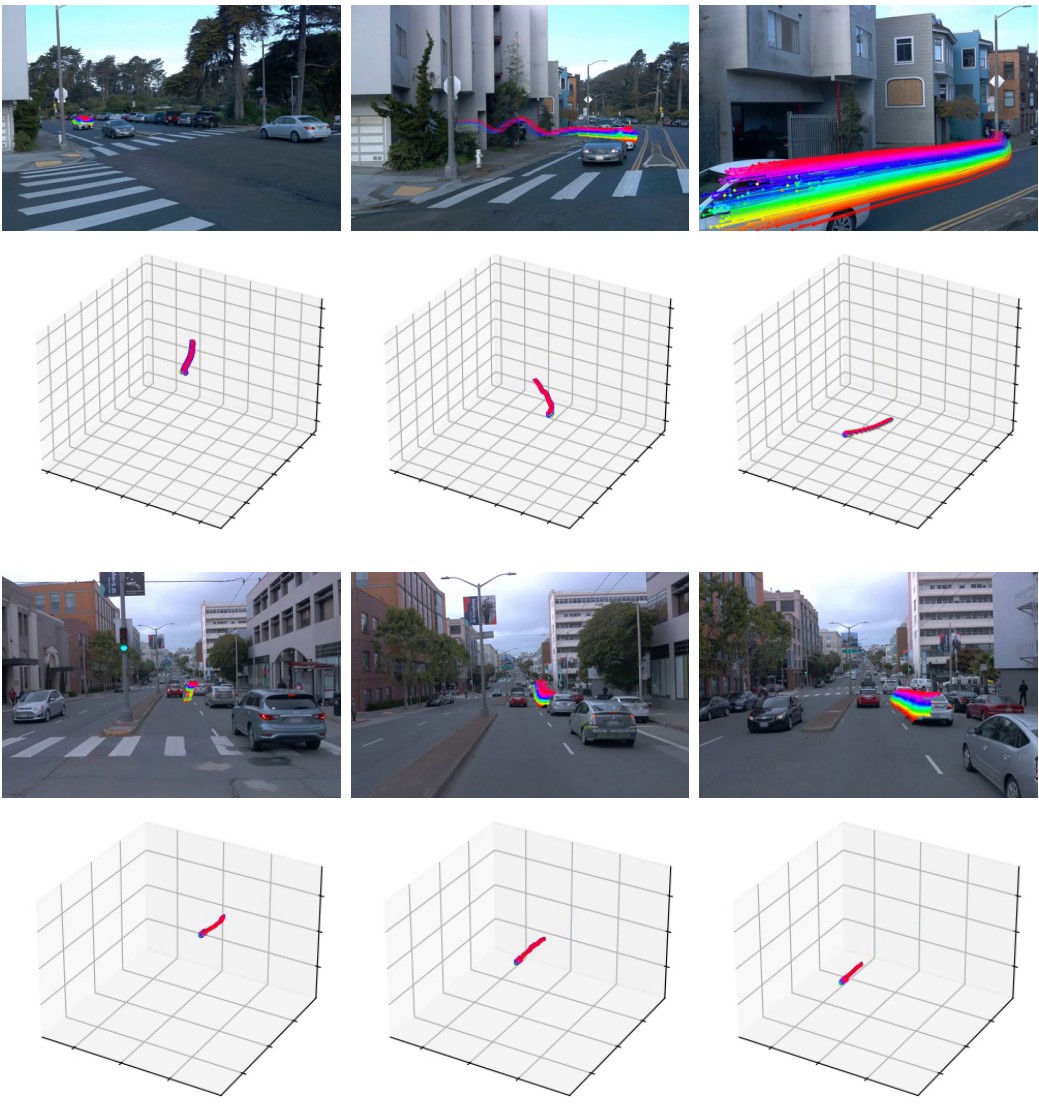

Figure 5: Random samples from DriveTrack subset in TAPVid-3D: on the top row, we visualize the point trajectories projected into the 2D video frame; on the bottom row, we visualize the metric 3D point trajectories. For each video, we show 3 frames sampled at time step 30, 60 and 90.

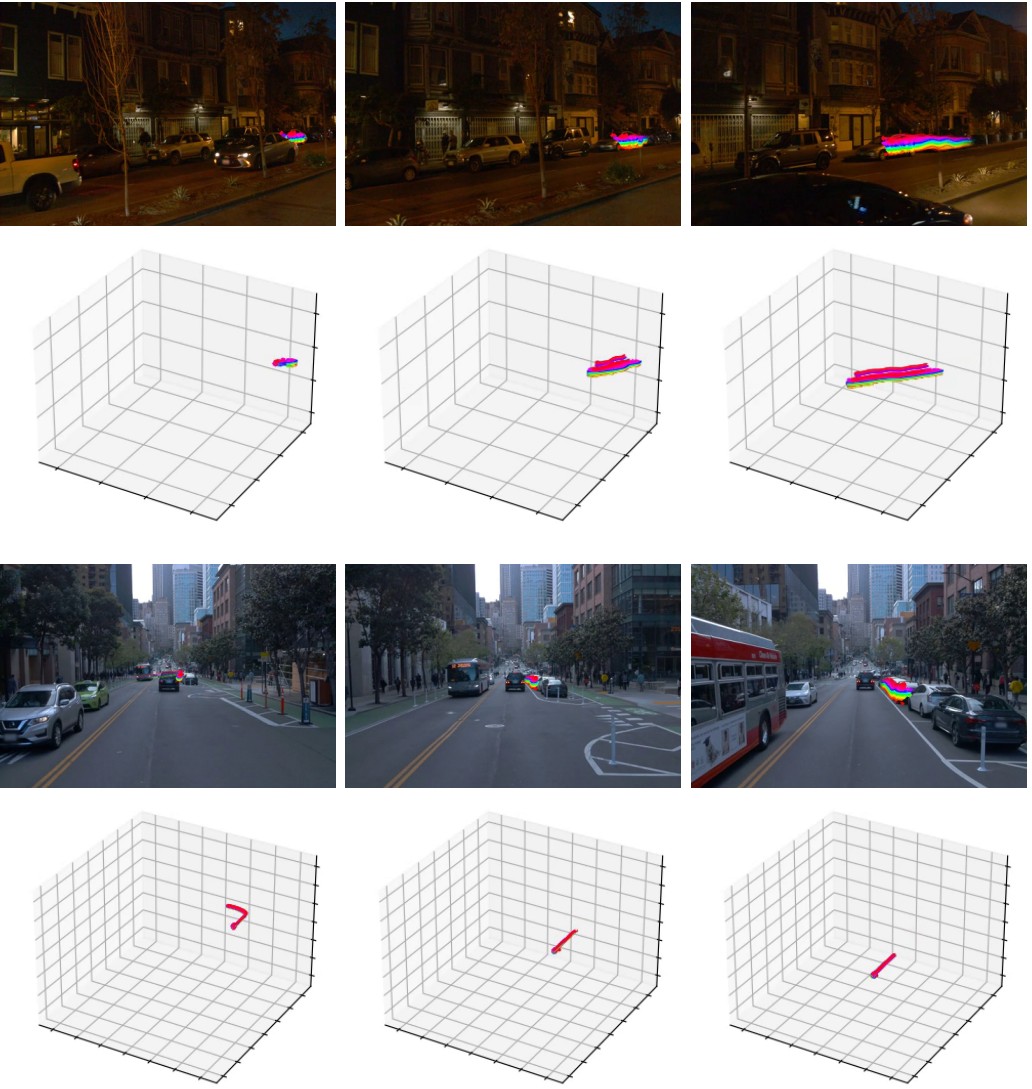

Figure 6: Random samples from DriveTrack subset in TAPVid-3D (cont'd.): on the top row, we visualize the point trajectories projected into the 2D video frame; on the bottom row, we visualize the metric 3D point trajectories. For each video, we show 3 frames sampled at time step 30, 60 and 90.

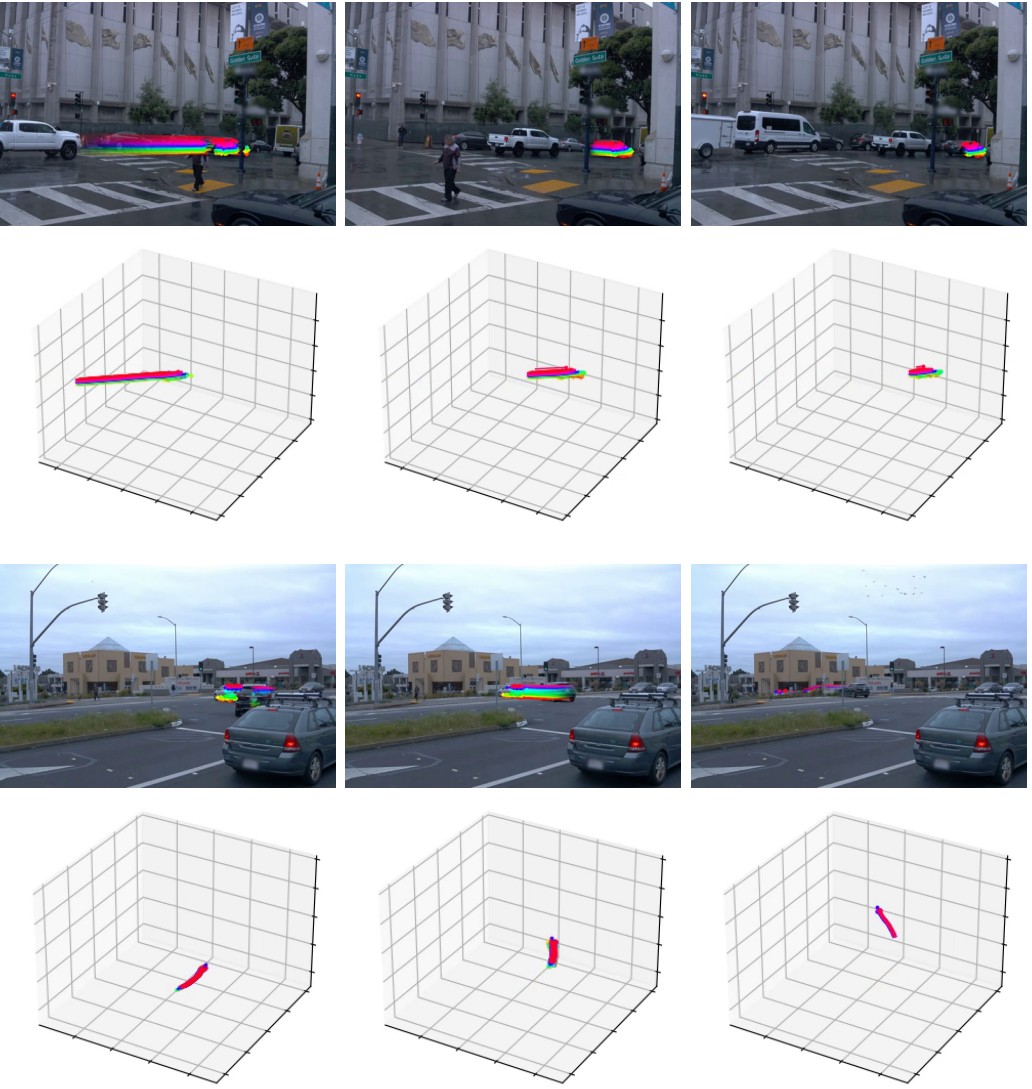

Figure 7: Random samples from DriveTrack subset in TAPVid-3D (cont'd.): on the top row, we visualize the point trajectories projected into the 2D video frame; on the bottom row, we visualize the metric 3D point trajectories. For each video, we show 3 frames sampled at time step 30, 60 and 90.

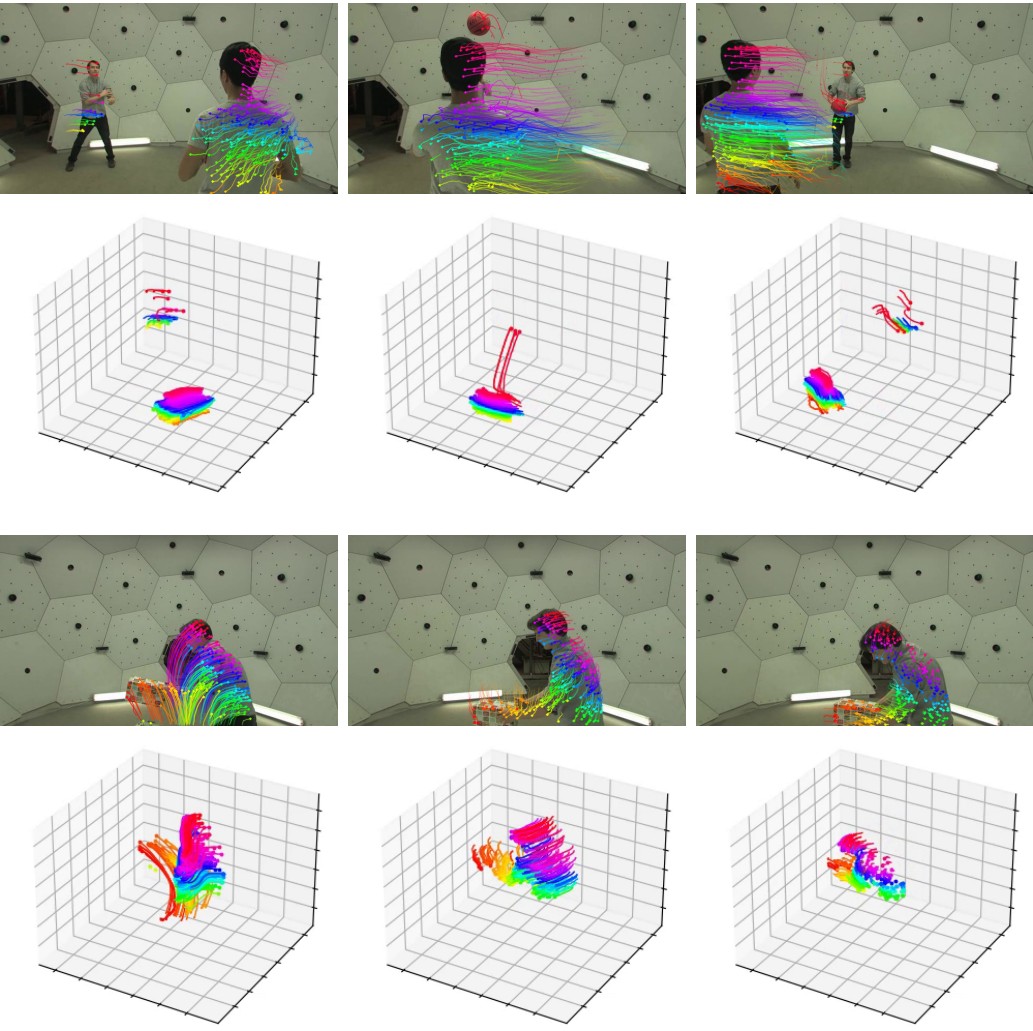

Figure 8: Random samples from Panoptic Studio subset in TAPVid-3D: on the top row, we visualize the point trajectories projected into the 2D video frame; on the bottom row, we visualize the metric 3D point trajectories. For each video, we show 3 frames sampled at time step 30, 60 and 90.

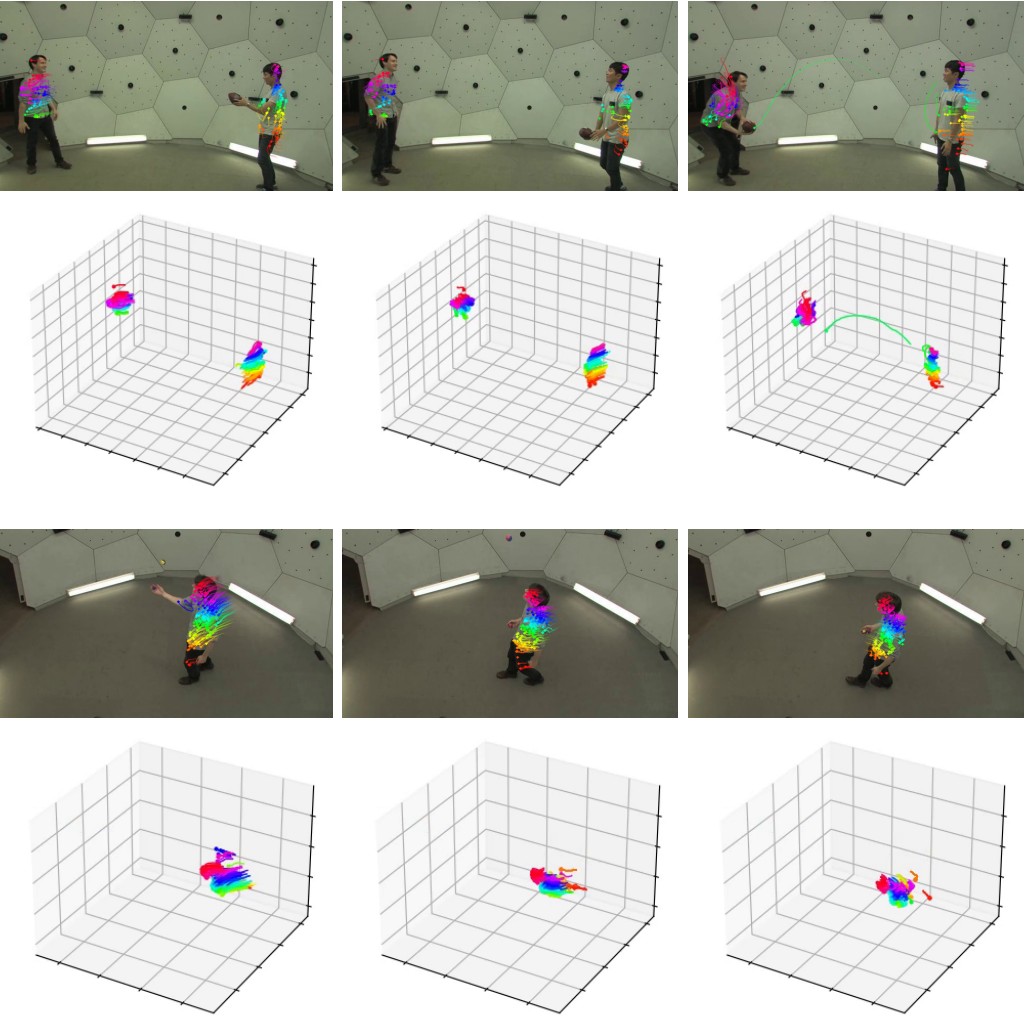

Figure 9: Random samples from Panoptic Studio subset in TAPVid-3D (cont'd.): on the top row, we visualize the point trajectories projected into the 2D video frame; on the bottom row, we visualize the metric 3D point trajectories. For each video, we show 3 frames sampled at time step 30, 60 and 90.

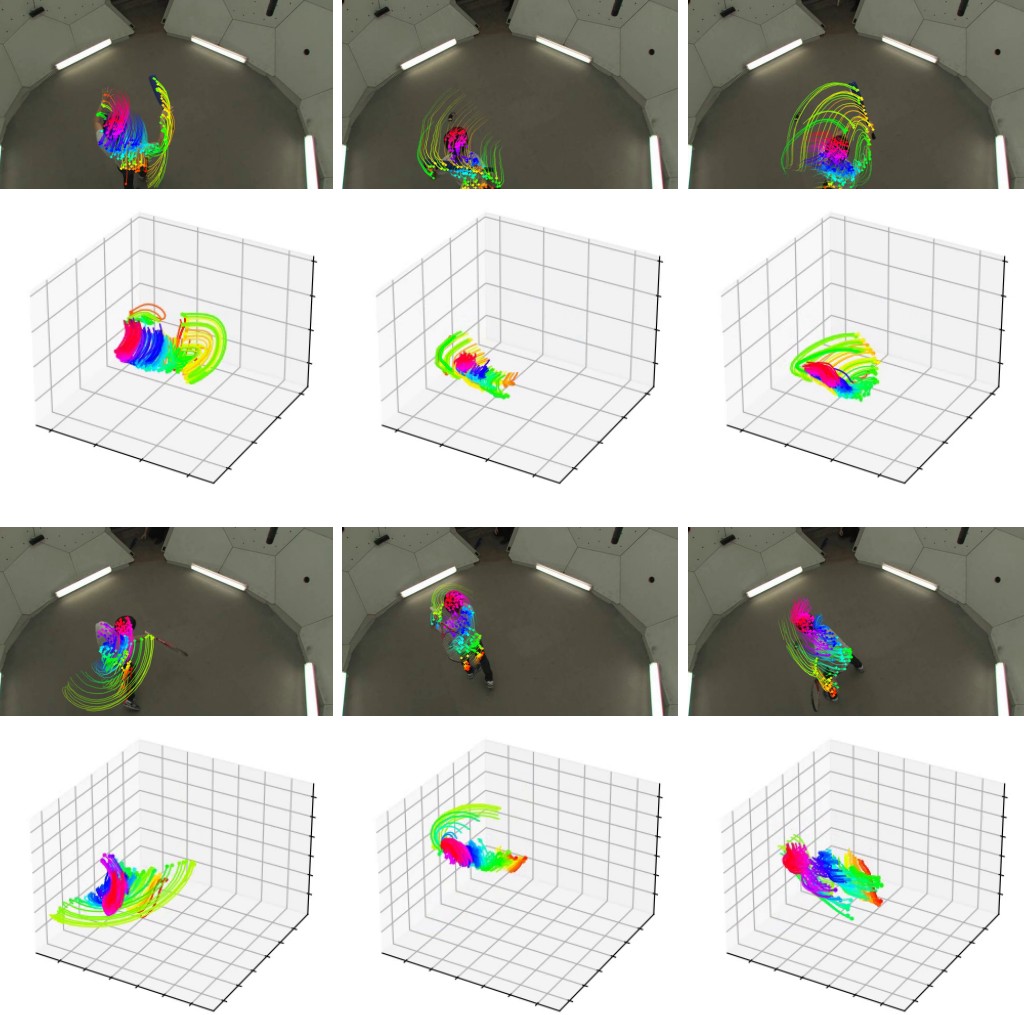

Figure 10: Random samples from Panoptic Studio subset in TAPVid-3D (cont'd.): on the top row, we visualize the point trajectories projected into the 2D video frame; on the bottom row, we visualize the metric 3D point trajectories. For each video, we show 3 frames sampled at time step 30, 60 and 90.