# OpenReview forum: "TAPVid-3D: A Benchmark for Tracking Any Point in 3D"
_NeurIPS.cc/2024/Datasets_and_Benchmarks_Track — NeurIPS 2024 Track Datasets and Benchmarks Poster_

### Official Review · Reviewer_oGa3 · 2024-06-20
**Review of TAPVid-3D**

**Rating:** 6
**Confidence:** 4
**Clarity:** Yes

**Review:**

Pros:
1. The benchmark provided in the paper can be beneficial for evaluating the performance of methods for 3D point tracking.
2. The data comes from different sources and includes multiple scenarios, which can increse the diversity.
3. The adoption of multiple ways to construct the pseudo-GT of the 3D trajectory is motivating.

Cons:
1. There is not enough analysis on the quality of the pseudo-GT of the 3D tracking. In the supplementary demo video, it appears that none of the scenarios are perfect. More statistics or human judgment on the percentage of correctness is needed.
2. Only the annotation pipeline (pretrained dynamic 3D Gaussian) in Panoptic Scene can deal with non-rigid objects. The strategies applied in the other two sources cannot be applied to deformable objects, which limits the diversity of the data.
3. The query point sampling is not very balanced, especially in the DriveTrack data source.

**Strengths:**

The proposed benchmark can be useful to the community for evaluating the performance of different point tracking problems. Additionally, it can further motivate other work on how to construct better datasets or annotation pipelines to achieve real GT 3D point tracking.

**Additional Feedback:**

The paper demonstrates that the data for the benchmark is designed for training. What are the main challenges blocking the scalability of the data construction pipeline or using existing data for training?

**Correctness:**

Yes, the benchmark is constructed reasonably, and the evaluation and experiments are understandable.

**Documentation:**

Yes

**Ethics:**

No ethical concerns

**Limitations:**

Yes, the authors mention that the provided dataset can inheirt some limititions from TAP and monocular depth.

**Opportunities For Improvement:**

There is not enough analysis on the quality of the pseudo-GT of the 3D tracking. In the supplementary demo video, it appears that none of the scenarios are perfect. More statistics or human judgment on the percentage of correctness is needed. It will be better if the authors can provide more manual analysis on the quality of the pseudo-GT.

**Relation To Prior Work:**

Yes

**Summary And Contributions:**

The paper introduces a benchmark for the task of tracking any point in 3D. The authors design a 3D tracking auto-annotation and filtering pipeline to create a dataset with over 4000 clips from three data sources, including the egocentric view, the auto-driving scenario, and third-view data. They additionally extend the evaluation metric in TAP-2D to 3D to provide an understanding of the performances of existing baselines.

---

> ### Author Rebuttal · Authors · 2024-08-15
>
> We thank the reviewer for the constructive feedback, positive comments about how our benchmark will be beneficial for the tracking community, and the diversity of the dataset.
>
> > More statistics or human judgment on the percentage of correctness is needed.
>
> We provide here additional details about the human quality assessment mentioned in section 3.4 (L229) of the main paper. Using four human annotators, we manually analyzed and conducted analysis on the tracks for a random sample of 150 videos (the minival split), covering approximately 66000+ trajectories. Annotators were provided for both the 2D projection of the 3D trajectories onto the original video, and also the 3D plot of the trajectories, and asked to identify and tally the number of incorrect trajectories. We find that nearly all trajectories (>99%) are able to pass human visual inspection. A breakdown of results by video source is shown below:
>
> | Data Source         | % of Trajectories Passing Human Verification |
> | ----------------------   | ------------------------------------------------------------ |
> | Aria Digital Twin   | 99.31%                                                              |
> | Panoptic Studio    | 98.79%                                                             |
> | DriveTrack            | 99.45%                                                             |
> |    |  |
> | **Overall**                  | **99.22%**                                                             |
>
> We hope this provides insight into the benchmark, and provides a quantitative answer to the reviewer's question.
>
> > Only the annotation pipeline in Panoptic Scene (dynamic 3D Gaussian) can deal with non-rigid objects. The strategies applied in the other two sources cannot be applied to deformable objects, which limits the diversity of the data.
>
> We appreciate the feedback, and indeed, we recognize that each of our splits has different characteristics: indoors vs outdoors, moving camera vs static camera, rigid vs non-rigid objects. However, we believe these characteristics are complementary, and combined, make the TAPVid3D benchmark a comprehensive evaluation of 3D point tracking.
>
> > The query point sampling is not very balanced (in DriveTrack)
>
> Query point sampling in Panoptic Studio and DriveTrack is largely focused on moving objects, which in our experience, is where most of the downstream use cases of visual tracking are actually used. Therefore, we build an evaluation benchmark that focuses on these moving objects. Nonetheless, to round out the benchmark we include the dense query point sampling in the Aria Digital Twins split, so the benchmark is able to test performance on background query points as well.
>
> But overall, we take this comment and the reviewer's previous comment as a great suggestion on areas in which to further develop, and something we will keep in mind for a future iteration of the dataset.

---

> > ### Comment · Reviewer_oGa3 · 2024-08-27
> >
> > Thanks for the responses from the authors and I will maintain my score.

---

### Official Review · Reviewer_yv9g · 2024-07-19
**Useful Benchmark for Relatively New 3D Task**

**Rating:** 7
**Confidence:** 3
**Correctness:** Yes, the claims made in this paper se…
**Clarity:** Yes, this paper is very well written …

**Review:**

Since this task is relatively new, it makes sense that a robust benchmark is needed for future methods designed for this task. The baseline evaluations seem thorough and the methods for data collection are documented. The paper is clear and well-organized, and the progression of the paper makes sense. As mentioned, since there is a clear need for such a benchmark, I can confidently state that this work is original and significant. The stated applications of this task, which can be accelerated using this benchmark, include autonomous driving and robotic manipulation, which I as others in 3D vision would agree are very important tasks. The quantitative baselines provided in this paper will be very useful to future works in this area. However, I would like to see a more in-depth documentation of the data collection process, well enough for one to reproduce this process. One intrinsic limitation of this work is that it is a dataset-dependent benchmark, rather than a standalone metric that can be used with any dataset, or collection of data, for this task.

**Strengths:**

To reiterate, this paper provides a benchmark for a nascent task in this field that is much needed. Again, the task at hand is a useful one with many applications, and I believe its development will be accelerated with the presence of a robust, trustworthy benchmark. The dataset seems well-motivated and reasonable in design, and the examples provided are promising. The mathematical rigor of the description of the dataset, particularly in Sections 3.1-3.3 and 3.5, are comprehensive, albeit slightly involved.

**Additional Feedback:**

None.

**Documentation:**

There is no code or data yet available, but the authors do state that they will be made available in the future.

**Ethics:**

No.

**Limitations:**

Yes! As a credit to the authors, they include a Limitations section that states that there are some domains not covered by this dataset. Furthermore, I appreciate that they acknowledge potential inaccuracies with respect to automatic annotations. The ethical considerations are spelled out, and I agree with them, or lack thereof.

**Opportunities For Improvement:**

Any major opportunity for improvement lies in the fact that while this work provides a solid dataset+benchmark, there is perhaps still room for more comprehensive metrics in this field that can be used with any dataset. Any other opportunities for improvement are minor:

There could perhaps be more scenes from dynamic interactions and extreme conditions, although this is perhaps more apt for a separate dataset.

**Relation To Prior Work:**

Yes, this paper describes the need for this benchmark in this task, in that there are no existing benchmarks for this nascent task.

**Summary And Contributions:**

This paper introduces a new benchmark called TAPVid-3D which evaluates the tracking "any point" in 3D space from real-world videos. The paper mentions that many benchmarks exist for tracking any point in 2D, but there is none for 3D. The benchmark sources data from three different datasets to collect over 4000 real-world video clips. These clips cover a gamut of settings, including various scenes, motions, objects, etc. In addition, this paper introduces new Jaccard-based metrics that are suited for evaluating methods on complex 3D scenarios. This metric is used to evaluate existing SOTA models for tracking any point in 3D, and provides benchmarks for future models to be evaluated on. The authors mention that important applications of this task include robotic manipulation, video editing, and others.

---

> ### Author Rebuttal · Authors · 2024-08-15
>
> Thank you for your helpful feedback and appreciation for the work. In response to the two points that the reviewer brought up:
>
> > there is perhaps still room for more comprehensive metrics in this field that can be used with any dataset.
>
> We wish to clarify that we did not create “a dataset-dependent benchmark, rather than a standalone metric”.  We believe that our 3D Average Jaccard (3D-AJ) metric can be used for *any* dataset that contains ground truth point tracks along with visibility flags, as it only depends on the x/y/z/visibility values for point tracks.  Indeed, there are no prior real-world datasets that currently provide this sort of annotations, but we believe that synthetic data, as well as our proposed approaches (1: converting pose estimates from motion capture to tracks, 2: using 4D reconstruction methods from multi-view, 3: using depth sensing hardware [e.g. LIDAR) together with pose estimates) can be applied to many other domains to create new datasets. We hope our metric will be applied to those future datasets as well. We will update the language in the paper to emphasize that the metric can be used independently of the dataset, as a way to measure performance on any future 3D TAP dataset.
>
> > There could perhaps be more scenes from dynamic interactions and extreme conditions, although this is perhaps more apt for a separate dataset.
>
> This is a great suggestion, and something we will keep in mind for a future iteration of the dataset. Obtaining 3D point tracking data usually requires complex multi-view camera streams or multi-sensor setup (which we leverage in this release), which somewhat limits the complexity of scenes and environments, particularly for outdoor scenes. We will give some more thought on how best to measure tracking in new, more complex dynamic scenes for a future version of the dataset.

---

> > ### Comment · Reviewer_yv9g · 2024-08-28
> > **Response**
> >
> > Thank you for the clarification, I will keep my score of 7 as it is a good work.

---

### Official Review · Reviewer_nHmX · 2024-07-24
**Evaluation dataset for long term 3D point tracking**

**Rating:** 6
**Confidence:** 3
**Correctness:** Yes.
**Clarity:** Yes.

**Review:**

This work address a relevant field that is growing fast in popularity, so the efforts in evaluating the methods in a more fairer and precise way are relevant. The quality of the dataset, in terms of presentation and usefulness are high.

**Strengths:**

Authors did a good work realizing the cavities of existing datasets in the field and fill them with a better packed benchmark. The dataset contains variated scenarios (indoor and outdoor), under different conditions (moving rigid objects as cars or fully non-rigid ones as humans), recorded with a moving camera. This makes it more complete than the previous datasets targeting the same problem of 3D point tracking.

**Additional Feedback:**

None.

**Documentation:**

Yes.

**Ethics:**

No.

**Limitations:**

Yes.

**Opportunities For Improvement:**

By the way the pseudo ground truth labels are generated (estimated monocular depth + 2D point tracking, Lidar, etc.), it is impossible to reach certain point of quality. Authors know this and correctly redirect the use of this benchmark for evaluation rather than training learning based methods. A more accurate way of creating this labels would be appreciated to be able to trust more the labels, but authors are naturally limited. Despite the intrinsic limitations in the creation of this kind of datasets, this work solves some cavities of the current benchmarks.

**Relation To Prior Work:**

Yes.

**Summary And Contributions:**

Authors release dataset targeting the evaluation of 3D point trackers. This benchmark comes from the process of existing datasets that were not offering this kind of 3D point tracking labels. The relevance of this field is gaining in popularity, and this dataset is more complete than existing ones, offering wider flexibility.

---

> ### Author Rebuttal · Authors · 2024-08-15
>
> Thank you for the feedback, and recognition of the benchmark's novelty and diversity. The reviewer brought up one main point to which we would like to respond:
>
> >  "the pseudo ground truth labels are generated (estimated monocular depth + 2D point tracking, Lidar, etc.), it is impossible to reach certain point of quality"
>
> We would like to clarify that "monocular depth" (eg. ZoeDepth) and "2D point tracking" (eg. BootsTAPIR) are only used for computing current baseline performance, and are not employed for constructing the ground truth trajectories.
>
> Instead, for Aria Digital Twin, the 3D ground truth comes from aligned synthetic rendered scenes, which are aligned to the ground-truth videos [41]. For DriveTrack, it comes from 3D LiDAR points which are propagated using the 3D bounding boxes derived from LiDAR, and for Panoptic Studio, it comes from dynamic 3D reconstructions obtained from 27 fixed-pose cameras surrounding the scene.
>
> The use of synthetic environments (ADT), LiDAR data (DriveTrack) or multi-view (Panoptic Studio) to obtain 3D trajectories is, while imperfect, significantly superior to what can currently be obtained from a single video sequence, which we believe provides a strong validation signal to improve single-view algorithms.
>
> Despite this, we agree that ground truth will not be perfect, as is the case of every TAP dataset containing real-world videos like TAP-Vid (for which human annotation introduces errors in the trajectories). To further assess the quality of our generated trajectories, we conduct a visual inspection of the generated trajectories. These results are discussed in the response to R4 (oGa3).

---

### Official Review · Reviewer_XvtV · 2024-07-24
**Great motivation and experimental demonstration**

**Rating:** 8
**Confidence:** 5
**Correctness:** Yes.
**Clarity:** Well-written.

**Review:**

The motivation of this paper is very clear, and their baseline estimation is solid. Their dataset covers various scenarios, with full 3D information available. The inclusion of 3D data allows for more accurate pixel-level occlusion and motion tracking since 3D information provides a better understanding of complex occlusions and motions. By handling three different datasets in real-world scenarios, the dataset is suitable for a variety of applications. For each dataset, the authors propose effective pre-processing and post-processing techniques to enhance dataset quality. While there are areas that could be improved, the overall quality is sufficiently high for acceptance.

**Strengths:**

The authors have provided thorough details to reproduce the process. They have clearly documented every aspect of their dataset creation and baseline evaluation, ensuring that others can accurately replicate their work. This includes comprehensive descriptions of the data collection, annotation procedures, and the metrics used for evaluation, which together contribute to the reliability and reproducibility of their research.

The paper is exceptionally well-organized and contains readable assets that enhance understanding. The authors have structured the content clearly, making it easy to follow their methodology and findings. For example, Table 1 is particularly well-designed, allowing readers to quickly grasp the advantages of using their dataset. This clarity in presentation ensures that the paper is accessible to a broad audience, including those who may not be specialists in the field.

The motivation for this research is articulated very clearly. Understanding the underlying 3D structure and the movements of 3D points is a crucial problem in computer vision, yet existing benchmarks were insufficient. The authors address this gap by introducing a comprehensive dataset specifically designed for 3D point tracking. This dataset will be instrumental in advancing the development of new methods for tracking 3D points, thereby contributing significantly to the field.

**Additional Feedback:**

The dataset will make a significant contribution to the vision community, providing valuable resources for advancing 3D point tracking research. With minor improvements in the paper, its overall quality will be enhanced, making the findings even more impactful and accessible to researchers in the field.

Some of concerns are clearly reserved after the rebuttal. I have decided to maintain my score to the original score.

**Documentation:**

They have sufficiently provided the details.

**Ethics:**

No ethical problems.

**Limitations:**

No limitations found for this dataset.

**Opportunities For Improvement:**

Table 2 is located very far from the main section that refers to it. This placement can disrupt the flow of reading and make it difficult for readers to connect the content with the relevant data. It would be much better if Table 2 were positioned closer to the section that discusses it, ensuring that readers can easily access and understand the table in the context of the accompanying text.

Table 3 does not include a sufficiently diverse range of baselines. To enhance the robustness and comprehensiveness of the experiments, I suggest adding more recent baselines, such as Geowizard and DepthAnything. Including these newer models will provide a richer set of comparisons and give a clearer picture of how the proposed method performs relative to the current state of the art.

It would be better if the captions for Table 4 and Table 5 were more self-descriptive. Currently, the metrics 3D-AJ, APD, and OA are not immediately understandable to readers encountering the tables for the first time. If the captions included a brief description or the full names of each metric, it would significantly improve readability and comprehension. This additional information would help readers quickly grasp the significance of the metrics and the results presented in the tables.

**Relation To Prior Work:**

Yes.

**Summary And Contributions:**

The paper introduces TAPVid-3D, a new benchmark for long-range 3D point tracking, addressing a gap in benchmarks for three-dimensional tracking tasks. Unlike 2D point tracking, which has several benchmarks like TAPVid-DAVIS, 3D tracking lacked a dedicated benchmark until now. TAPVid-3D consists of over 4,000 real-world videos sourced from three different datasets, showcasing a variety of object types, motion patterns, and both indoor and outdoor environments. The benchmark uses metrics based on the Jaccard index, adapted to handle challenges like ambiguous depth, occlusions, and the need for spatio-temporal smoothness in tracking. A significant portion of video annotations has been manually verified to ensure accuracy. To establish a performance baseline, existing tracking models were tested on this new benchmark. TAPVid-3D aims to enhance the understanding of 3D motion and surface deformation using monocular video, providing a valuable resource for future research and development in 3D point tracking.

---

> ### Author Rebuttal · Authors · 2024-08-15
>
> Thank you for the detailed review and constructive comments. We were really happy to see that you found the benchmark useful. We also appreciate the positive feedback on the paper clarity, and robustness of the methods and baselines.
>
> We will update the final manuscript to incorporate all the provided suggestions. In particular:
>
> 1. We have repositioned Table 2 to be closer to the corresponding text.
>
> 2. We have made the captions for Table 4 and Table 5 more self-descriptive, and briefly described the meaning of the 3D-AJ, APD, and OA metrics, adding the following: "Occlusion Accuracy (OA) measures accuracy of point visibility classification, Average Position Error within Delta measures the point position error across all trajectories in a video, and the TAP-3D headline metric, 3D Average Jaccard (3D-AJ), combines these two as a measure of overall performance on the TAP-3D task, factoring in both visibility and position errors."
>
> 3. We have added DepthAnything V2 and GeoWizard (together with our strongest 2D point tracker, BootsTAPIR) to Table 3. The updated table is also attached in the accompanying PDF, for your convenience. We find both of these models do not perform significantly better than the previous baselines. Looking at samples, we suspect that single frame monocular depth estimators, like Depth Anything V2, are bottlenecked by temporally inconsistent (flickering) depth predictions across the video, which causes errors in the overall point trajectory.
>
> We will add these changes to the final version of the paper.

---

> > ### Comment · Reviewer_XvtV · 2024-08-29
> > **Thanks for your rebuttal. Some of concerns are fully addressed.**
> >
> > Some of concerns are clearly reserved after the rebuttal. I have decided to maintain my score to the original score.

---

### Author Rebuttal · Authors · 2024-08-15

Thank you to all four reviewers for their thoughtful feedback and overall appreciation for our work. We were happy to find that all reviewers noted the usefulness of the work, mentioning how it would be "instrumental in advancing the development of new methods" (R1), "a valuable resource for future research" (R1), and how the benchmark is "much needed" (R3) with "high usefulness" (R2). We also appreciate the positive feedback on the paper itself, describing it as "exceptionally well-organized" (R1), "clear" (R3, R1), and "comprehensive" (R3). Finally, many reviewers commented positively on both the diversity of the dataset ("more complete than the previous datasets" [R2], "covers a gamut of settings" [R2], "includes different sources and multiple scenarios" [R4]), and the strength of the baseline models ("baseline evaluations seem thorough" [R3], and "baseline estimation is solid" [R1]).

We have responded to reviewer-specific comments as replies in each thread.

R1: XvtV, R2: nHmX, R3: yv9g, R4: oGa3.

---

### Decision · Program_Chairs · 2024-09-26

**Decision:**

Accept (Poster)

**Comment:**

The paper introduces TAPVid-3D, a new benchmark for long-range 3D point tracking, addressing a gap in benchmarks for three-dimensional tracking tasks. All reviewers rated it positively and believe it will benefit the community. The AC recommends acceptance.